# Incorporating Domain Knowledge in VAE Learning via Exponential Dissimilarity-Dispersion Family

## Abstract

Variational autoencoder (VAE) is a prominent generative model that has been actively applied to various unsupervised learning tasks such as representation learning. Despite its representational capability, VAEs with the commonly adopted Gaussian settings typically suffer from performance degradation in generative modeling for high-dimensional natural data, which is often caused by their excessively limited model family. In this paper, we introduce the exponential dissimilarity-dispersion family (EDDF), a novel distribution family that includes a dissimilarity function and a global dispersion parameter. A decoder with this distribution family induces arbitrary dissimilarity functions as the reconstruction loss of the evidence lower bound (ELBO) objective, where the model leverages domain knowledge through this dissimilarity function. For VAEs with EDDF decoders, we also propose an ELBO optimization method that implicitly approximates the stochastic gradient of the normalizing constant using log-expected dissimilarity. Empirical evaluations of the generative performance show the effectiveness of our model family and proposed method in the vision domain, indicating that the effect of dissimilarity determines the criteria of representational informativeness.

## 1 Introduction

VAE (Kingma & Welling, 2014; Rezende et al., 2014) is an influential generative model extensively applied to unsupervised learning tasks such as probabilistic modeling and representation learning. The applications of VAE models are attributable to their ability to learn a latent coordinate system of the underlying data manifold (Goodfellow et al., 2016). VAEs learn a latent variable to form a low-dimensional representation of high-dimensional data and bidirectionally infer them by their encoder–decoder network structure. Such a *latent representation* is extensively applied as an informative and concise data representation to several tasks such as real-world recognition (Ha & Schmidhuber, 2018; Higgins et al., 2017b; Li et al., 2023), abstract reasoning (van Steenkiste et al., 2019), and knowledge discovery (Liu et al., 2020; Takahashi et al., 2022).

The framework of VAEs has a solid theoretical foundation based on variational inference for generative modeling (Kingma & Welling, 2014; Rezende et al., 2014); however, in naturally observed high-dimensional data, VAEs with the commonly used settings practically suffer from performance degradation, such as insufficient reconstruction fidelity and generation naturalness (Larsen et al., 2016; Hou et al., 2019). The contributory factors of such shortcomings primarily fall into the following two viewpoints as for the choice of the decoder distribution: **(i)** simple and oft-used decoder families (*e.g.*, isotropic Gaussian) represent several excessively strong assumptions about data variables (Burda et al., 2016; Kingma et al., 2016); **(ii)** ignoring the decoder dispersion and normalizing constant causes a balancing problem between reconstruction and regularization (Rezende & Viola, 2018; Lin et al., 2019; Rybkin et al., 2021). These two viewpoints imply that an appropriate choice of training implementation enables VAE models to practically perform as the variational inference framework theoretically promises.

**Viewpoint (i).** The decoder distribution is conventionally set to several well-known distributions to induce simple and easy-to-implement reconstruction losses, *e.g.*, the isotropic Gaussian represents the sum of squared error (SSE) loss. These losses yield simple and closed-form objectives enabling

easy implementation and analysis; however, such a simple and general model occasionally conflicts with the domain-specific property of the observed data for effectively modeling the underlying latent manifold of real-world observations. For example, local patches of natural images form a manifold (Carlsson et al., 2008; Carlsson, 2009), suggesting that the isotropic Gaussian implying pixel-wise independence is incompatible with the domain knowledge on the vision domain. To overcome these issues with such well-known distributions, several types of advanced loss functions for the reconstruction term have been introduced, *e.g.*, an adversarially learned metric (Larsen et al., 2016), a classifier-based metric (Hou et al., 2019), and domain-specific guidance (Ding et al., 2020). Although they improve performance in specific tasks, such additional or ad-hoc mechanisms could disrupt the benefit and generality of the original VAE framework, such as stable training.

**Viewpoint (ii).** Simply using an arbitrary function for the reconstruction term of the ELBO objective results in a balancing problem due to the scale mismatch against the regularization term. $\beta$-VAE (Higgins et al., 2017a) has assigned a hyperparameter weight $\beta$ to the regularization term for balancing against the reconstruction, which has been later re-interpreted as the rate-distortion tradeoff (Alemi et al., 2018). Too low $\beta$ causes performance degradation in generation and representation learning due to posterior-prior discrepancies (Dai & Wipf, 2019), and too high $\beta$ causes posterior collapse and over-smoothed latent representations (Burgess et al., 2017; Fu et al., 2019; Takida et al., 2022). Although the performance of $\beta$-VAE models is thus sensitive to the $\beta$ value due to the tradeoff, several studies (Rezende & Viola, 2018; Lucas et al., 2019) have addressed this balancing problem by introducing a global dispersion parameter (Lucas et al., 2019; Lin et al., 2019). The optimal dispersion is analytically tractable using the expected reconstruction error (Rezende & Viola, 2018; Lin et al., 2019; Rybkin et al., 2021), improving the VAE performance even in decoders based on simple distributions such as isotropic Gaussian (Rybkin et al., 2021). Albeit its generality, the dispersion parameter of VAE decoders is primarily applied only to well-known and relatively simple distributions in the literature, such as isotropic Gaussian distributions (Lucas et al., 2019; Rybkin et al., 2021) and the exponential dispersion family (Sicks et al., 2021).

In this paper, we propose an extended VAE framework by introducing a distribution family EDDF, and an ELBO optimization method based on the approximation of the EDDF normalizing constant. EDDF decoders leverage domain-specific dissimilarity functions that acts as a strong induced bias for natural high-dimensional data. This dissimilarity function is designed based on knowledge in a particular domain. The proposed optimization method provides a tractable approximated objective for VAEs with EDDF decoders in a near-optimal dispersion parameter. The contributions of this study are as follows:

- We propose a novel distribution family EDDF for the VAE decoder, which uses a dissimilarity function established on domain knowledge as an inductive bias for natural high-dimensional data.

- We propose an approximated algorithm using a log-expected dissimilarity loss function to optimize VAEs with EDDF decoders.

- We empirically validate an approximated algorithm, showing that the trained autoencoder estimates optimal dispersion, and reveal that EDDF decoders in VAEs effectively utilize domain knowledge.

## 2 VARIATIONAL AUTOENCODER (VAE)

VAE is a powerful approach to the generative modeling of high-dimensional natural data using autoencoding-like neural networks (NNs). Given an $N$-sized set of independent and identically distributed data points $\mathcal{D} = \{\mathbf{x}^{(i)}\}_{i=1}^N$ for $M$-dimensional data variables $\mathbf{x} \sim p_{\mathcal{D}}(\mathbf{x})$, we assume an $L$-dimensional ($L \ll M$) latent variable $\mathbf{z}$ distributed on the prior $\pi(\mathbf{z})$. Because the motivation of this assumption is the manifold hypothesis (Carlsson et al., 2008; Carlsson, 2009; Bengio et al., 2013), it induces a formulation in which the data variable $\mathbf{x}$ is supported on a differentiable data manifold $\mathcal{X} \subseteq \mathbb{R}^M$ with a local coordinate system referred to as a latent space $\mathcal{Z} \subseteq \mathbb{R}^L$. To model these assumptions, we consider a VAE model (Kingma & Welling, 2014) with a negative ELBO

objective $\mathcal{L}_{\mathrm{ELBO}}$ as

$$\begin{aligned}
\mathcal{L}_{\mathrm{ELBO}} := -&\mathbb{E}_{q_\phi(\mathbf{x},\mathbf{z})}\left[\log p_{\boldsymbol{\theta}}(\mathbf{x}|\mathbf{z})\right] \\
&+ \mathbb{E}_{p_{\mathcal{D}}(\mathbf{x})}\left[D_{KL}\left(q_\phi(\mathbf{z}|\mathbf{x})\|\pi(\mathbf{z})\right)\right],
\end{aligned} \tag{1}$$

whose minimization can be interpreted as a probabilistic autoencoding using an encoder $q_\phi(\mathbf{z}|\mathbf{x})$ with parameters $\phi$ and a decoder $p_{\boldsymbol{\theta}}(\mathbf{x}|\mathbf{z})$ with parameters $\boldsymbol{\theta}$. This yields a generative model $p_{\boldsymbol{\theta}}(\mathbf{x},\mathbf{z})$ and an inference model $q_\phi(\mathbf{x},\mathbf{z})$ that can be mutually approximated, as shown by

$$p_{\boldsymbol{\theta}}(\mathbf{x},\mathbf{z}) = p_{\boldsymbol{\theta}}(\mathbf{x}|\mathbf{z})\pi(\mathbf{z}), \tag{2}$$

$$q_\phi(\mathbf{x},\mathbf{z}) = q_\phi(\mathbf{z}|\mathbf{x})p_{\mathcal{D}}(\mathbf{x}), \tag{3}$$

where $\pi(\mathbf{z})$ denotes a prior distribution of the latent variables $\mathbf{z}$. A typical choice of the assumptions in the literature is Gaussian models parameterized by NNs (Kingma & Welling, 2014; Alemi et al., 2018; Lucas et al., 2019; Sicks et al., 2021). In such settings, the generative process comprises a standard Gaussian prior $\pi(\mathbf{z}) = \mathcal{N}(\mathbf{0}, \mathbf{I}_L)$ and the Gaussian decoder parameterized by a NN $\hat{\mathbf{x}}_{\boldsymbol{\theta}} : \mathcal{Z} \to \mathcal{X}$ as

$$p_{\boldsymbol{\theta}}(\mathbf{x}|\mathbf{z}) = \mathcal{N}(\mathbf{x}|\hat{\mathbf{x}}_{\boldsymbol{\theta}}(\mathbf{z}), \gamma\mathbf{I}_M), \tag{4}$$

where a global dispersion parameter $\gamma > 0$ is postulated as in Rybkin *et al.* (Rybkin et al., 2021). Because the inverted generation process $p_{\boldsymbol{\theta}}(\mathbf{z}|\mathbf{x})$ is intractable in this case, it is approximated with a diagonal Gaussian encoder $q_\phi(\mathbf{z}|\mathbf{x})$ parameterized by NNs $\boldsymbol{\mu}_\phi : \mathcal{X} \to \mathcal{Z}$ and $\boldsymbol{\sigma}_\phi : \mathcal{X} \to \mathbb{R}_{>0}^L$ as

$$q_\phi(\mathbf{z}|\mathbf{x}) = \mathcal{N}(\mathbf{z}|\boldsymbol{\mu}_\phi(\mathbf{x}), \mathrm{diag}(\boldsymbol{\sigma}_\phi^2(\mathbf{x}))), \tag{5}$$

where $\mathrm{diag}(\cdot)$ denotes a diagonal matrix with the input vector as its diagonal elements.

The variational inference of Eq. (1) using the generative and inference models produces an interpretation as a variational autoencoding, where the first term of Eq. (1) is referred to as a reconstruction loss $\mathcal{L}_{\mathrm{rec}}$, and the second term of Eq. (1) is interpreted as a regularization loss $\mathcal{L}_{\mathrm{reg}}$, as follows:

$$\mathcal{L}_{\mathrm{rec}} := -\mathbb{E}_{q_\phi(\mathbf{x},\mathbf{z})}\left[\log p_{\boldsymbol{\theta}}(\mathbf{x}|\mathbf{z})\right], \tag{6}$$

$$\mathcal{L}_{\mathrm{reg}} := \mathbb{E}_{p_{\mathcal{D}}(\mathbf{x})}\left[D_{KL}\left(q_\phi(\mathbf{z}|\mathbf{x})\|\pi(\mathbf{z})\right)\right]. \tag{7}$$

Into the regularization $\mathcal{L}_{\mathrm{reg}}$, $\beta$-VAE (Higgins et al., 2017a) introduces a weighted objective with a weight hyperparameter $\beta$ as follows:

$$\mathcal{L}_\beta = \mathcal{L}_{\mathrm{rec}} + \beta\mathcal{L}_{\mathrm{reg}}. \tag{8}$$

Here, we can balance these two losses by calibrating the $\beta$ value. However, its performance is sensitive to $\beta$ whose optimal value depends on the dataset $\mathcal{D}$ (Lucas et al., 2019; Fil et al., 2021; Nakagawa et al., 2023), requiring intense hyperparameter tuning and resource-consuming model selection (Locatello et al., 2019; Duan et al., 2020). As another approach to the calibration of the weight $\beta$, several studies have focused on a decoder dispersion that induces a trainable coefficient equivalent to the loss weighting of $\beta$-VAE (Rezende & Viola, 2018; Lucas et al., 2019; Lin et al., 2019). For example, the Gaussian case in Eq. (4) produces $\mathcal{L}_{\mathrm{rec}}$ with a square error as

$$\mathcal{L}_{\mathrm{rec}} = \mathbb{E}_{q_\phi(\mathbf{x},\mathbf{z})}\left[\frac{\|\mathbf{x} - \hat{\mathbf{x}}_{\boldsymbol{\theta}}(\mathbf{z})\|_2^2}{2\gamma} + \frac{M}{2}\log 2\pi\gamma\right], \tag{9}$$

where $\gamma$ is equivalent to the weight $\beta$ if $\gamma$ is constant with regard to the model parameters $(\boldsymbol{\theta}, \phi)$.

## 3 EXPONENTIAL DISSIMILARITY-DISPERSION FAMILY (EDDF)

To design a probabilistic decoder with a strong inductive bias based on domain knowledge, we define a distribution with a dissimilarity function $d : \mathcal{X}^2 \to \mathbb{R}_{\geq 0}$, a location parameter $\mathbf{m} \in \mathcal{X}$, and a dispersion parameter $\gamma \in \mathbb{R}_{>0}$ as

$$f_d(\mathbf{x}|\mathbf{m}, \gamma) = \exp\left[-\frac{d(\mathbf{x}, \mathbf{m})}{\gamma} + C_d(\mathbf{m}, \gamma)\right], \tag{10}$$

where $C_d(\mathbf{m}, \gamma)$ denotes the normalizing constant, and the dissimilarity function $d$ satisfies the following conditions:

$$\forall\mathbf{x}, \mathbf{m} \in \mathcal{X}, \quad \mathbf{x} = \mathbf{m} \implies d(\mathbf{x}, \mathbf{m}) = 0, \tag{11}$$

$$\forall\mathbf{x}, \mathbf{m} \in \mathcal{X}, \quad d(\mathbf{x}, \mathbf{m}) \geq 0, \tag{12}$$

$$\forall\mathbf{x}, \mathbf{m} \in \mathcal{X}, \quad \nabla_{\mathbf{m}}d(\mathbf{x}, \mathbf{m}) \text{ exists}, \tag{13}$$

Table 1: An overview of well-known distribution families that can be interpreted as a subset of EDDF. The parameters are re-interpreted in the form of the location $\mathbf{m}$ and the dispersion $\gamma$. Although the location parameter $\mathbf{m}$ becomes the mode $\arg\max_{\mathbf{m}} f_d(\mathbf{x}|\mathbf{m}, \gamma)$, the dispersion parameter $\gamma$ does not necessarily represent the variance in these cases. $x_i$ and $m_i$ denote the $i$-th elements of $\mathbf{x}$ and $\mathbf{m}$, respectively.

| Distribution $f_d(\mathbf{x}|\mathbf{m}, \gamma)$ | Domain $\mathrm{dom}(d(\cdot, \mathbf{m}))$ | Dispersion $\gamma$ | Dissimilarity $d$ |
|---|---|---|---|
| $\mathcal{N}(\mathbf{m}, \gamma\mathbf{I}_M)$ | $\mathbb{R}^M$ | $\gamma > 0$ | $1/2 \, \|\mathbf{x} - \mathbf{m}\|_2^2$ |
| $\mathrm{Laplace}(\mathbf{x}|\mathbf{m}, \gamma)$ | $\mathbb{R}^M$ | $\gamma > 0$ | $1/2 \, \|\mathbf{x} - \mathbf{m}\|_1$ |
| $\mathrm{vMF}(\mathbf{x}|\mathbf{m}, 1/\gamma)$ | Unit Sphere $S^{M-1}$ | $\gamma > 0$ | $-\mathbf{x}^\top\mathbf{m}$ |
| $\mathrm{Bernoulli}(\mathbf{x}|\mathbf{m})$ | $\{0, 1\}^M$ | fixed $\gamma = 1$ | $-\sum_{i=1}^M [x_i \log m_i + (1 - x_i)\log(1 - m_i)]$ |
| $\mathrm{Gamma}(\mathbf{x}|1 + \frac{\mathbf{m}}{\gamma}, \gamma)$ | $(0, \infty)^M$ | $\gamma > 0$ | $\sum_{i=1}^M [x_i - m_i \log x_i]$ |

where the first and second conditions require that the location $\mathbf{m}$ produces the mode, and the third one indicates the differentiable dissimilarity for optimization and analysis. Note that the dissimilarity $d$ is not necessarily a metric because the symmetry and triangle inequality axioms are not required in this definition. The normalizing constant $C_d(\mathbf{m}, \gamma)$ enforces the measure $f_d$ to produce a value of 1 for the entire support, as expressed below:

$$C_d(\mathbf{m}, \gamma) = -\log \int_{\mathrm{dom}(d(\cdot, \mathbf{m}))} \exp\left[-\frac{d(\mathbf{x}, \mathbf{m})}{\gamma}\right] d\mathbf{x}, \tag{14}$$

where $\mathrm{dom}$ denotes the domain of a function. A finite normalizing constant results in a condition

$$\int_{\mathrm{dom}(d(\cdot, \mathbf{m}))} \exp\left[-\frac{d(\mathbf{x}, \mathbf{m})}{\gamma}\right] d\mathbf{x} < \infty, \tag{15}$$

which implies that the dissimilarity function $d$ must be an unbounded function $d(\mathbf{x}, \mathbf{m}) = \infty$ as $\|\mathbf{x}\| \to \infty$ if the domain $\mathrm{dom}(d(\cdot, \mathbf{m}))$ is unbounded.

### 3.1 Relations to Well-Known Distributions

The probability density function of EDDF distributions in Eq. (10) resembles the exponential family, and several distributions of the exponential family are included in the EDDF. Thus, the EDDF distributions include several commonly adopted settings using well-known distributions, and the details are presented in Table 1. For example, using the mean squared error (MSE) or SSE as a dissimilarity $d_{\ell^2}(\mathbf{x}, \mathbf{m}) := 1/2\|\mathbf{x} - \mathbf{m}\|_2^2$ produces the family of isotropic Gaussian distributions $f_{d_{\ell^2}}(\mathbf{x}|\mathbf{m}, \gamma) = \mathcal{N}(\mathbf{x}|\mathbf{m}, \gamma\mathbf{I}_M)$. Although the exponential family requires the dot-product form of the variable and parameters, EDDF is less restricted in this respect, allowing more flexible interaction between the data $\mathbf{x}$ and the location $\mathbf{m}$, *e.g.*, using perceptual metrics. Here, we highlight that exponential family distributions can be characterized using Bregman divergences, a concept detailed in "Information Geometry and Its Applications", Section 2.7 (Amari, 2016). This framework provides a rigorous basis for the variational inference techniques used in our approach.

## 4 VAE Optimization with EDDF Decoders via Log-Expected Dissimilarity Loss

First, to introduce the EDDF distributions to the VAE decoder, we define the decoder model with an analytic dissimilarity function $d$ as

$$p_{\boldsymbol{\theta}}(\mathbf{x}|\mathbf{z}) = f_d(\mathbf{x}|\hat{\mathbf{x}}_{\boldsymbol{\theta}}(\mathbf{z}), \gamma). \tag{16}$$

To maximize the ELBO in this model, we first propose a theoretical approach that this optimization is approximately possible in a similar manner to the previously studied analytical optimal (Rybkin et al., 2021; Lin et al., 2019). The optimal dispersion $\gamma^*$ is induced by the decoder model, as shown below:

$$\gamma^* \in \arg\max_{\gamma \in \mathbb{R}_{>0}} \mathcal{L}_{\mathrm{ELBO}}(\boldsymbol{\theta}, \boldsymbol{\phi}, \gamma)$$

$$= \arg\max_{\gamma \in \mathbb{R}_{>0}} \mathcal{L}_{\mathrm{rec}}(\boldsymbol{\theta}, \boldsymbol{\phi}, \gamma) + \mathcal{L}_{\mathrm{reg}}(\boldsymbol{\phi})$$

$$= \arg\max_{\gamma \in \mathbb{R}_{>0}} \mathcal{L}_{\mathrm{rec}}(\boldsymbol{\theta}, \boldsymbol{\phi}, \gamma), \tag{17}$$

where the reconstruction loss $\mathcal{L}_{\text{rec}}$ is induced by the EDDF in Eq. (16) as

$$\mathcal{L}_{\text{rec}}(\boldsymbol{\theta}, \boldsymbol{\phi}, \gamma) = \mathbb{E}_{q_{\boldsymbol{\phi}}(\mathbf{x}, \mathbf{z})} \left[ \frac{d(\mathbf{x}, \hat{\mathbf{x}}_{\boldsymbol{\theta}}(\mathbf{z}))}{\gamma} - C_d(\hat{\mathbf{x}}_{\boldsymbol{\theta}}(\mathbf{z}), \gamma) \right]. \tag{18}$$

To compute the normalizing term including an intractable integral in Eq. (14) approximately, we expand the dissimilarity $d$ at the location parameter $\mathbf{m}$ as

$$d(\mathbf{x}, \mathbf{m}) = \frac{1}{2}(\mathbf{x} - \mathbf{m})^{\mathsf{T}} \mathbf{H_m}(\mathbf{x} - \mathbf{m}) + f(\mathbf{x} - \mathbf{m}), \tag{19}$$

where $\mathbf{H_m}$ denotes the Hessian matrix at $\mathbf{m}$ defined as $\mathbf{H_m} := \nabla_{\mathbf{m}} \nabla_{\mathbf{m}}^{\mathsf{T}} d(\mathbf{x}, \mathbf{m})$, and $f(\mathbf{x})$ denotes the remainder term $f(\mathbf{x}) = O(\|\mathbf{x}\|^3)$ as $\|\mathbf{x}\| \to +0$. This expansion produces an asymptotic normalizing constant

$$C_d(\mathbf{m}, \gamma) \leq \underbrace{\frac{1}{2} \log \det \frac{\mathbf{H_m}}{2\pi\gamma}}_{=: \tilde{C}_d(\mathbf{m}, \gamma)} + O(\gamma^{1/2}) \qquad \text{as } \gamma \to +0. \tag{20}$$

The asymptotic normalizing constant $\tilde{C}_d(\mathbf{m}, \gamma)$ gives an asymptotic reconstruction loss $\tilde{\mathcal{L}}_{\text{rec}}(\boldsymbol{\theta}, \boldsymbol{\phi}, \gamma)$ as

$$\mathbb{E}_{q_{\boldsymbol{\phi}}(\mathbf{x}, \mathbf{z})} \left[ \frac{d(\mathbf{x}, \hat{\mathbf{x}}_{\boldsymbol{\theta}}(\mathbf{z}))}{\gamma} - \tilde{C}_d(\hat{\mathbf{x}}_{\boldsymbol{\theta}}(\mathbf{z}), \gamma) \right]$$

$$= \frac{1}{\gamma} \mathbb{E}_{q_{\boldsymbol{\phi}}(\mathbf{x}, \mathbf{z})} \left[ d(\mathbf{x}, \hat{\mathbf{x}}_{\boldsymbol{\theta}}(\mathbf{z})) \right] + \frac{M}{2} \log(2\pi\gamma)$$

$$- \frac{1}{2} \mathbb{E}_{q_{\boldsymbol{\phi}}(\mathbf{x}, \mathbf{z})} \left[ \log \det \mathbf{H}_{\hat{\mathbf{x}}_{\boldsymbol{\theta}}(\mathbf{z})} \right]. \tag{21}$$

Considering that the last term is independent of the dispersion $\gamma$, this asymptotic reconstruction loss produces an approximated optimal dispersion $\tilde{\gamma}^*$ as

$$\tilde{\gamma}^* = \frac{2D_{\boldsymbol{\theta}, \boldsymbol{\phi}}}{M}, \tag{22}$$

where $D_{\boldsymbol{\theta}, \boldsymbol{\phi}}$ denotes the expected reconstruction dissimilarity $D_{\boldsymbol{\theta}, \boldsymbol{\phi}} := \mathbb{E}_{q_{\boldsymbol{\phi}}(\mathbf{x}, \mathbf{z})} [d(\mathbf{x}, \hat{\mathbf{x}}_{\boldsymbol{\theta}}(\mathbf{z}))]$. Because the asymptotic reconstruction loss $\tilde{\mathcal{L}}_{\text{rec}}$ converges to the original $\mathcal{L}_{\text{rec}}$ if $\gamma \to +0$, the approximated optimal $\tilde{\gamma}^*$ asymptotically becomes an unbiased estimator of the exact optimal dispersion $\gamma^*$ if $D_{\boldsymbol{\theta}, \boldsymbol{\phi}} \to +0$. This setting $D_{\boldsymbol{\theta}, \boldsymbol{\phi}} \approx 0$ implies that near-perfect reconstruction is feasible and practical in high-capacity VAEs, as suggested in the literature (Rybkin et al., 2021) that $D_{\boldsymbol{\theta}, \boldsymbol{\phi}} \approx 0.006 \ll 1$ is selected for the best generative performance in the SVHN dataset (Netzer et al., 2011). This "well-trained autoencoder" assumption leads to a further easy-to-implement approximation; the expected log-determinant Hessian term of Eq. (21) can be assumed to be constant in this case because well-trained autoencoders produce better reconstruction $\mathbf{x} \approx \hat{\mathbf{x}}_{\boldsymbol{\theta}}(\mathbf{z})$ in which the data points $\mathbf{x}$ lie close to the minimizing point of the dissimilarity $d(\cdot, \hat{\mathbf{x}}_{\boldsymbol{\theta}}(\mathbf{z}))$. This assumption and the above consequences in Eqs. (21) and (22) produce the gradients of the asymptotic reconstruction loss

$$\nabla_{\boldsymbol{\theta}, \boldsymbol{\phi}} \tilde{\mathcal{L}}_{\text{rec}} \Big|_{\gamma = \tilde{\gamma}^*} = \frac{M}{2} \frac{\nabla_{\boldsymbol{\theta}, \boldsymbol{\phi}} \mathbb{E}_{q_{\boldsymbol{\phi}}(\mathbf{x}, \mathbf{z})} [d(\mathbf{x}, \hat{\mathbf{x}}_{\boldsymbol{\theta}}(\mathbf{z}))]}{\mathbb{E}_{q_{\boldsymbol{\phi}}(\mathbf{x}, \mathbf{z})} [d(\mathbf{x}, \hat{\mathbf{x}}_{\boldsymbol{\theta}}(\mathbf{z}))]}$$

$$= \nabla_{\boldsymbol{\theta}, \boldsymbol{\phi}} \frac{M}{2} \log \mathbb{E}_{q_{\boldsymbol{\phi}}(\mathbf{x}, \mathbf{z})} [d(\mathbf{x}, \hat{\mathbf{x}}_{\boldsymbol{\theta}}(\mathbf{z}))]. \tag{23}$$

Finally, we introduce a log-expected reconstruction loss

$$\mathcal{L}_{\text{rec}}^{\asymp} = \frac{M}{2} \log \mathbb{E}_{q_{\boldsymbol{\phi}}(\mathbf{x}, \mathbf{z})} [d(\mathbf{x}, \hat{\mathbf{x}}_{\boldsymbol{\theta}}(\mathbf{z}))], \tag{24}$$

which is identical to $\tilde{\mathcal{L}}_{\text{rec}}$ up to the constant and produces the gradients with regard to the encoder-decoder parameters $(\boldsymbol{\theta}, \boldsymbol{\phi})$ in Eq. (23). Because the training process of VAEs is implemented using the Monte Carlo sampling method (Kingma & Welling, 2014), we can implement the log-expected reconstruction loss $\mathcal{L}_{\text{rec}}^{\asymp}$ without adding any additional trainable parameters. For example, in a differentiable programming framework PyTorch (Paszke et al., 2019), this loss $\mathcal{L}_{\text{rec}}^{\asymp}$ can be implemented as `dissimilarities.mean().log()`. The entire negative ELBO objective, particularly under the condition where $\gamma$ is sufficiently small ($\gamma \ll 1$), simplifies to

$$\mathcal{L}_{\text{ELBO}} \big|_{\gamma = \gamma^*} \approx \mathcal{L}_{\text{rec}}^{\asymp} + \mathcal{L}_{\text{reg}}, \tag{25}$$

where $\mathcal{L}_{\text{reg}}$ is based on the isotropic Gaussian assumption as detailed in Eq. (7). The constant term, not contributing to the gradients, is omitted in the optimization process.

Table 2: An overview of the visual datasets used: for those without a provided validation set (denoted by *), we split the training dataset 90% for training and 10% for validation. #Train, #Valid, and #Test sizes are noted, and the "Aligned" column indicates if image positions are aligned (✓) or not (–).

| Dataset | #Train | #Valid | #Test | Aligned |
|---|---|---|---|---|
| MNIST (LeCun et al., 1998) * | 54,000 | 6,000 | 10,000 | ✓ |
| SVHN (Netzer et al., 2011) * | 65,931 | 7,326 | 26,032 | ✓ |
| CelebA (Liu et al., 2015) | 162,770 | 19,867 | 19,962 | ✓ |
| CIFAR-10 (Krizhevsky & Hinton, 2009) * | 45,000 | 5,000 | 10,000 | – |
| Food-101 (Bossard et al., 2014) * | 68,175 | 7,575 | 25,250 | – |

## 5 RELATED WORKS

Numerous studies have concentrated on rectifying practical issues in the VAE framework, despite its theoretical foundations (Kviman et al., 2023; Liévin et al., 2023; Lin et al., 2023). Balancing the reconstruction and regularization losses within the rate–distortion tradeoff has been a classic focus (Alemi et al., 2018; Tishby et al., 1999; Tschannen et al., 2018). This tradeoff allows the weighting of the reconstruction loss to produce informative latent variables about data and the weighting of the regularization loss to offer concise latent variables for disentanglement (Higgins et al., 2017a).

Several studies have addressed the $\beta$-VAE balancing problem by proposing algorithms to tune the weight parameter $\beta$ or the equivalent dispersion parameter $\gamma$, driven by the dependency of the optimal $\beta$ value on datasets and model settings. A significant example is GECO (Rezende & Viola, 2018), which transforms the negative ELBO objective of VAEs into a constrained optimization problem using Lagrange multipliers for reconstruction loss. The GECO algorithm employs a reconstruction upper bound as a hyperparameter to balance losses, with the tuning effect observed through reconstruction quality (De Boom et al., 2021); however, manual selection of the balancing parameter remains an issue. Introducing trainable dispersion parameters offers another method to balance the ELBO objective's terms (Lucas et al., 2019; Lin et al., 2019; Rybkin et al., 2021). This approach optimizes dispersion parameters within the log-likelihood framework for generative modeling $p_{\boldsymbol{\theta}}(\mathbf{x}) \approx p_{\mathcal{D}}(\mathbf{x})$. Rybkin et al. (2021) specifically explored this design, utilizing a global dispersion parameter $\gamma$. This parameter, an analytically solved scalar value based on reconstruction loss, leads to stable training and easily implemented models that enhance VAE decoder variances (Kingma & Welling, 2014). However, the analysis's applicability is restricted to general natural data due to reliance on isotropic Gaussian decoders.

Structural similarity (SSIM) (Wang et al., 2004) is a widely-used index measuring image similarity, differing from MSE loss in its consideration of natural image structure. Various indices, including one by Reisenhofer et al. (2018) using the Haar wavelet, have been proposed for enhanced performance (Wang & Li, 2011; Sheikh & Bovik, 2006; Zhang et al., 2011; Zhang & Li, 2012; Xue et al., 2014; Zhang et al., 2014; Balanov et al., 2015; Reisenhofer et al., 2018; Ziaei Nafchi et al., 2016). Several similarity indices based on deep convolutional neural networks (CNNs) have been proposed to measure the "perceptual" similarity between images (Gatys et al., 2016; Zhang et al., 2018; Prashnani et al., 2018; Ding et al., 2022). As deep learning (DL)-based image classifiers have demonstrated high capability (Krizhevsky et al., 2012), their hidden features have been actively studied for an application as a visual similarity measure. The hidden features of deep CNNs provide a perceptually preferable metric (Zhang et al., 2018). Many methods use deep visual features as the measure of the discrepancy between images in various ways, *e.g.*, using shallow and deep layers separately for evaluating the style and content of images (Gatys et al., 2016). In the VAE literature (Larsen et al., 2016; Hou et al., 2019), dissimilarity functions in the DL framework have been used as a reconstruction loss for practical use in forms such as adversarially trained metrics (Larsen et al., 2016) and a metric based on pretrained visual features (Hou et al., 2019). Deep CNNs-based similarity indices measure "perceptual" similarity between images (Gatys et al., 2016; Zhang et al., 2018; Prashnani et al., 2018; Ding et al., 2022). DL classifiers' hidden features are studied for visual similarity, providing a perceptually preferable metric (Zhang et al., 2018). Various methods use these features in different ways, such as evaluating style and content (Gatys et al., 2016). In VAE literature, dissimilarity functions in the DL framework are used for practical reconstruction loss, including adversarially trained metrics (Larsen et al., 2016).

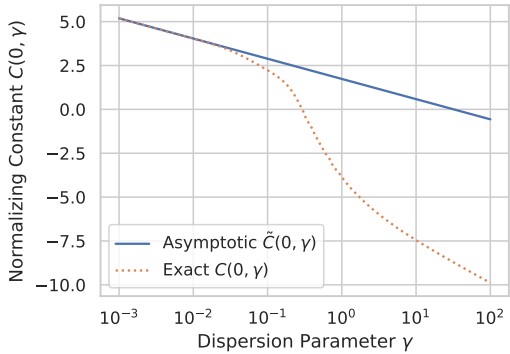

Figure 1: A plot for the exact $C_d(0, \gamma)$ (dashed, orange) and the asymptotic $\tilde{C}_d(0, \gamma)$ (solid, blue) in a toy dissimilarity function $d(x, 0) = 1 - \exp\left(-x^2\right) + \sin^2\left(10x\right)$ in a domain $\text{dom}(d(\cdot, 0)) = [-5, 5]$. The asymptotic normalizing constant converges to the exact value as $\gamma \to 0$.

## 6 EXPERIMENTS

We investigated the validity and effectiveness of the proposed distribution family EDDF for VAE decoders in the vision domain. First, we study the validity of the asymptotic normalizing constant of EDDF distributions. Second, for Viewpoint (i), we investigate the effectiveness of the log-expected reconstruction loss in estimating the optimal dispersion. Third, for Viewpoint (ii), we evaluated how the choice of the dissimilarity function $d$ affects VAE models with EDDF decoders.

### 6.1 SETTINGS

We provide empirical studies for the validation and benefit of our approximated VAE optimization method of EDDF decoders. We used small and simple convolutional NNs for all experimented VAE models. The encoder comprises four convolution layers and two fully-connected layers, whereas the decoder has the opposite architecture consisting of two fully-connected layers and four transposed convolution layers. For all models, the number of latent units $L$ is set to $L = 16$ in MNIST (LeCun et al., 1998), and $L = 64$ in relatively complex datasets (CelebA (Liu et al., 2015), SVHN (Netzer et al., 2011), Food-101 (Bossard et al., 2014), and CIFAR-10 (Krizhevsky & Hinton, 2009)). Following the benchmark in the literature (Dai & Wipf, 2019; Rybkin et al., 2021), we quantitatively investigated the VAE models in the visual domain using several image datasets as presented in Table 2. We evaluated the generative modeling of the experimented VAEs in the visual domain, using Fréchet Inception Distance (FID) (Heusel et al., 2017) and Kernel Inception Distance (KID) (Bińkowski et al., 2018).[1] All images used in the experiments were resized to $64 \times 64$ pixels. For CelebA (Liu et al., 2015), we cropped $144 \times 144$ pixels in the center of the original images to remove backgrounds. The implementation and evaluation of the models are based on PyTorch (Paszke et al., 2019) and PIQ (Kastryulin et al., 2022).

### 6.2 RESULTS

**Validity of Asymptotic Normalizing Constant.** To validate the asymptotic normalizing constant $\tilde{C}_d(\mathbf{m}, \gamma)$ introduced in Eq. (20), we investigated a one-dimensional toy model with a non-quadratic dissimilarity function. In this one-dimensional case, the data variable is a real scalar $\mathcal{X} = \mathbb{R}$, where the exact normalizing constant can be numerically computed. For simplicity, we adopt the zero value of the location parameter in this example, as presented in Fig. 1. The results of Fig. 1 illustrate the validity of the quadratic approximation even in a non-quadratic dissimilarity function, as the asymptotic normalizing constant $\tilde{C}_d(0, \gamma)$ converges to the exact $C_d(0, \gamma)$ in very small $\gamma \ll 1$. In a practical case, a run of the model with the log-expected reconstruction loss $\mathcal{L}_{\text{rec}}^{\approx}$ resulted in $\gamma = 0.095$, which we believe to be sufficiently small for the asymptotic approximation. These results suggest that the assumption $\gamma \ll 1$ is suitable in the training process of VAEs.

---

[1]Note that the exact ELBO values are not available because an intractable constant is ignored in Eq. (24).

Table 3: Comparison of different dispersion/weight tuning methods for VAEs. All the compared models were trained using isotropic Gaussian decoders, producing a simple MSE reconstruction loss. ↓ means that lower values represent better performance.

| Dispersion $\gamma$ / Weight $\beta$ | MNIST | | SVHN | | CelebA | | Food-101 | | CIFAR-10 | |
|---|---|---|---|---|---|---|---|---|---|---|
| | FID ↓ | KID ↓ | FID ↓ | KID ↓ | FID ↓ | KID ↓ | FID ↓ | KID ↓ | FID ↓ | KID ↓ |
| $\beta$-VAE, $\beta = 0.001$ | 95.17 | 0.1341 | 31.89 | 0.1138 | 65.51 | 0.0533 | 181.34 | 0.2558 | 87.61 | 0.1478 |
| $\beta$-VAE, $\beta = 0.01$ | 89.54 | 0.1146 | 24.46 | 0.1030 | 57.41 | 0.0498 | 154.32 | 0.2524 | 81.92 | 0.1493 |
| $\beta$-VAE, $\beta = 0.1$ | 70.75 | 0.0864 | **15.97** | 0.0897 | **48.98** | **0.0418** | 151.30 | 0.2436 | 72.89 | **0.1066** |
| $\beta$-VAE, $\beta = 1.0$ | 33.41 | **0.0656** | 20.52 | 0.1199 | 81.77 | 0.0450 | 189.24 | 0.2406 | 86.70 | 0.1317 |
| $\beta$-VAE, $\beta = 10.0$ | 34.68 | 0.1117 | 88.19 | 0.2028 | 179.59 | 0.0757 | 266.27 | 0.2386 | 176.89 | 0.2200 |
| $\beta$-VAE, $\beta = 100.0$ | 334.55 | 0.5140 | 204.84 | 0.5465 | 312.18 | 0.1973 | 373.23 | 0.5207 | 325.98 | 0.4093 |
| Trainable $\gamma$ | 75.54 | 0.0923 | 23.79 | 0.1079 | 58.40 | 0.0489 | 151.79 | 0.2444 | 74.62 | 0.1105 |
| GECO (Rezende & Viola, 2018) | 35.81 | 0.1191 | 38.97 | **0.0738** | 91.53 | 0.0556 | 185.20 | 0.2649 | 93.76 | 0.1039 |
| $\sigma$-VAE (Rybkin et al., 2021) | 82.56 | 0.1015 | 18.44 | 0.0952 | 53.66 | 0.0445 | 138.13 | 0.2298 | 73.61 | 0.1192 |
| $\tilde{\gamma}^*$ via $\mathcal{L}_{\text{rec}}^{\times}$ (Ours) | **27.57** | 0.0858 | 34.17 | 0.1159 | 60.19 | 0.0511 | **136.81** | **0.2412** | **65.38** | 0.1696 |

(a) Input Images

(b) MSE

(c) Content Score

(d) Style Score

(e) LPIPS

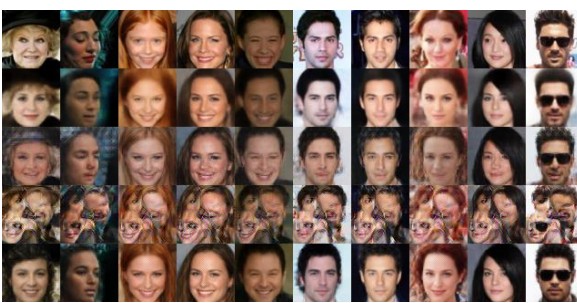

Figure 2: Qualitative comparison of reconstructed images using different dissimilarity functions, with columns corresponding to CelebA's test set data points (Liu et al., 2015). The top row shows ground-truth images, while subsequent rows display images reconstructed by EDDF decoders, with dissimilarity functions noted on the left.

**Effect of Log-Expected Dissimilarity Loss.** To investigate the effect of the log-expected dissimilarity loss in Eq. (24), we compared the generative performance of our EDDF-VAE with other dispersion calibration methods in Table 3. All models reported in this subsection use isotropic Gaussian decoders and EDDF with MSE to measure the effect of the implicit optimization of the dispersion parameter $\gamma$ by the log-expected dissimilarity loss. Although the manually tuned $\beta$ achieved high performance, the automatic tuning methods of dispersion $\gamma$ compared favorably on several datasets, particularly in two complex datasets Food-101 (Bossard et al., 2014) and CIFAR-10 (Krizhevsky & Hinton, 2009). In these results, the proposed method achieved performance similar to existing balance tuning methods, even in a very simple MSE dissimilarity.

**Selection of Dissimilarity Function.** To examine how the choice of dissimilarity function affects VAE models with EDDF decoders, we compared their reconstruction in Fig. 2 and generative performance in Table 4. In these experiments, we used the dissimilarity functions, as discussed in Section 5. The qualitative results of Fig. 2 suggest that the choice of dissimilarity indicates what type of information is considered informative for the data variable. Compared with other models, the reconstructed images of Style Score (Gatys et al., 2016) have completely distorted shapes of the faces even though the local structures of the faces are preserved (*e.g.*, the mouth-like object at the bottom left). The distortion of position and the preservation of semantic parts imply that the choice of the dissimilarity function determines what is preserved in the latent variable.

The quantitative results in Table 4 show that generative performance depends strongly on the choice of dissimilarity function $d$ and the domain of the dataset. Many hand-crafted dissimilarity functions produced NaN gradients that terminated the training process on all datasets, probably due to the function design being unsuitable for backpropagation (*e.g.*, taking the square root of zero). We tried to evaluate the functions on VIFp (Sheikh & Bovik, 2006), FSIM (Zhang et al., 2011), SR-SIM (Zhang & Li, 2012), GMSD (Xue et al., 2014), VSI (Zhang et al., 2014), and DSS (Balanov et al., 2015) in

Table 4: Quantitative comparison of EDDFs with different dissimilarity functions. – indicates that the training was terminated unsuccessfully due to the NaN gradient value, and ↓ denotes a metric in which lower values indicate better performance.

| Dissimilarity $d$ | MNIST FID ↓ | MNIST KID ↓ | SVHN FID ↓ | SVHN KID ↓ | CelebA FID ↓ | CelebA KID ↓ | Food-101 FID ↓ | Food-101 KID ↓ | CIFAR-10 FID ↓ | CIFAR-10 KID ↓ |
|---|---|---|---|---|---|---|---|---|---|---|
| MSE | 27.57 | 0.0858 | 34.17 | 0.1159 | 60.19 | 0.0511 | 136.81 | 0.2412 | 65.38 | 0.1696 |
| MAE | 22.26 | **0.0579** | 31.98 | 0.1133 | 52.89 | 0.0495 | 149.69 | 0.2826 | 58.53 | 0.1158 |
| Cross Entropy (CE) | 28.54 | 0.0813 | **15.91** | 0.1316 | 71.44 | 0.0430 | 188.46 | 0.2839 | 109.90 | 0.1838 |
| Cosine Similarity | 25.95 | 0.0748 | 35.81 | 0.1201 | 59.69 | 0.0483 | 156.33 | 0.3064 | 68.74 | 0.1965 |
| SSIM (Wang et al., 2004) | 22.74 | 0.0589 | 31.83 | **0.0834** | 40.17 | 0.0496 | 168.92 | 0.3650 | 69.38 | 0.1461 |
| IW-SSIM (Wang & Li, 2011) | 91.34 | 0.3244 | – | – | 217.41 | 0.3432 | – | – | 191.63 | 0.3532 |
| Content Score (Gatys et al., 2016) | 76.41 | 0.3278 | 152.47 | 0.3028 | 78.22 | 0.0869 | 199.02 | 0.2315 | 156.68 | 0.1806 |
| Style Score (Gatys et al., 2016) | 200.88 | 0.5764 | 163.07 | 0.2876 | 237.74 | 0.2274 | 157.26 | 0.2265 | 145.30 | 0.1986 |
| HaarPSI (Reisenhofer et al., 2018) | 43.85 | 0.2539 | 290.12 | 0.4736 | 302.17 | 0.3509 | 330.97 | 0.3917 | 246.36 | 0.3351 |
| LPIPS (Zhang et al., 2018) | 67.59 | 0.3235 | 47.87 | 0.3048 | 25.57 | 0.0806 | 97.96 | **0.1636** | **43.66** | 0.1480 |
| PieAPP (Prashnani et al., 2018) | 370.99 | 0.5389 | 201.68 | 0.5006 | 387.40 | 0.6130 | 421.78 | 0.5927 | 327.23 | 0.5070 |
| DISTS (Ding et al., 2022) | 27.72 | 0.2974 | 44.84 | 0.1855 | 29.50 | 0.0607 | **84.59** | 0.1813 | 60.10 | 0.0980 |
| MSE+LPIPS | **21.03** | 0.1911 | 30.00 | 0.2597 | 22.69 | 0.0389 | 107.88 | 0.1739 | 57.51 | 0.1051 |
| MAE+LPIPS | 21.71 | 0.1244 | 38.86 | 0.2804 | **20.36** | **0.0266** | 109.45 | 0.1681 | 64.73 | **0.0742** |

Table 5: Comparison of various VAE-based methods, indicated by a downward arrow for metrics where lower is better. Each method including ours was tuned using validation FID score. Selected dissimilarity functions for datasets MNIST, SVHN, CelebA, Food-101, and CIFAR-10 were MSE+LPIPS, CE, MAE+LPIPS, DISTS, and LPIPS, respectively (also see Table 4).

| Dispersion $\gamma$ / Weight $\beta$ | MNIST FID ↓ | MNIST KID ↓ | SVHN FID ↓ | SVHN KID ↓ | CelebA FID ↓ | CelebA KID ↓ | Food-101 FID ↓ | Food-101 KID ↓ | CIFAR-10 FID ↓ | CIFAR-10 KID ↓ |
|---|---|---|---|---|---|---|---|---|---|---|
| VAE-GAN (Larsen et al., 2016) | 40.63 | 0.2026 | 40.51 | 0.2332 | 34.86 | 0.0625 | 133.42 | 0.2389 | 64.40 | 0.1619 |
| 2-Stage VAE (Dai & Wipf, 2019) | 62.24 | 0.1546 | 42.32 | **0.0885** | 100.48 | 0.0796 | 262.40 | 0.3249 | 105.64 | 0.1421 |
| DFCVAE (Hou et al., 2019) | 97.30 | 0.3839 | 108.37 | 0.3042 | 61.36 | 0.0637 | 173.43 | 0.2317 | 217.57 | 0.2063 |
| Soft-IntroVAE (Daniel & Tamar, 2021) | 29.86 | 0.0996 | 22.81 | 0.1462 | 113.00 | 0.0584 | 200.81 | 0.2552 | 104.32 | 0.1210 |
| $\sigma$-VAE (Rybkin et al., 2021) | 82.56 | **0.1015** | 18.44 | 0.0952 | 53.66 | 0.0445 | 138.13 | 0.2298 | 73.61 | **0.1192** |
| Ours | **21.03** | 0.1911 | **15.91** | 0.1316 | **20.36** | **0.0266** | **84.59** | **0.1813** | **43.66** | 0.1480 |

addition to the methods in Table 4; however, all of them output the NaN gradient value. Although the DL-based dissimilarities in the lower part of Table 4 achieved high performance on relatively complex datasets (*e.g.*, CelebA), simple and classical dissimilarities such as MSE and cosine similarity achieved better results than the DL-based dissimilarities, suggesting that the required strength of the inductive biases depends on dataset complexity.

**Effectiveness of Proposed Method.** To investigate the effectiveness of our method, we quantitatively compared VAE based on EDDF decoders with existing extended VAE models in Table 5. The methods differ in terms of their training objectives and procedures, whereas their models share the identical architecture of encoder and decoder NNs. These quantitative results show that the proposed decoder family and optimization method compare favorably with existing VAE-based approaches, suggesting that it is effective to introduce domain-specific dissimilarity functions as an inductive bias in a probabilistically consistent manner.

## 7 Conclusion

We have proposed a novel probabilistic distribution family EDDF for VAE decoders and an approximated optimization method. The EDDF distributions with domain-specific dissimilarity functions work as strong inductive biases for complex high-dimensional data while preserving the probabilistic framework of VAEs. The experiments in the vision domain show the effect of our method. This investigation extends the frontiers of VAEs, a critical axis within the generative modeling paradigm, which also encompasses GANs and diffusion models. Our sophisticated application of exponential family distributions could have far-reaching implications, potentially enhancing the fidelity of sample synthesis and the granularity of diffusion process regulation in these complementary generative frameworks.

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

## A    THE PROOF OF EQUATION (20)

The second-order Taylor expansion in Eq. (19) induces the proof of Eq. (20).

*Proof.*

$$C_d(\mathbf{m}, \gamma)$$

$$= -\log \int_{\text{dom}(d(\cdot, \mathbf{m}))} \exp\left[-\frac{d(\mathbf{x}, \mathbf{m})}{\gamma}\right] d\mathbf{x} \tag{26}$$

$$= -\log \int_{\text{dom}(d(\cdot+\mathbf{m}, \mathbf{m}))} \exp\left[-\frac{\boldsymbol{\delta}^{\mathsf{T}} \mathbf{H}_{\mathbf{m}} \boldsymbol{\delta}}{2\gamma} + \frac{f(\boldsymbol{\delta})}{\gamma}\right] d\boldsymbol{\delta} \tag{27}$$

$$= -\log \mathbb{E}_{\mathcal{N}(\boldsymbol{\delta}|\gamma \mathbf{H}_{\mathbf{m}}^{-1})}\left[\exp\left(-\frac{f(\boldsymbol{\delta})}{\gamma} - \frac{1}{2}\log \det \frac{\mathbf{H}_{\mathbf{m}}}{2\pi\gamma}\right)\right] \tag{28}$$

where $\boldsymbol{\delta}$ denotes the deviation $\boldsymbol{\delta} := \mathbf{x} - \mathbf{m}$. Applying Jensen's inequality to this equation, as $\gamma \to +0$, we obtain

$$C_d(\mathbf{m}, \gamma) \leq \mathbb{E}_{\mathcal{N}(\boldsymbol{\delta}|\gamma \mathbf{H}_{\mathbf{m}}^{-1})}\left[\frac{f(\boldsymbol{\delta})}{\gamma} + \frac{1}{2}\log \det \frac{\mathbf{H}_{\mathbf{m}}}{2\pi\gamma}\right] \tag{29}$$

$$= \tilde{C}_d(\mathbf{m}, \gamma) + \mathbb{E}_{\mathcal{N}(\boldsymbol{\delta}|\gamma \mathbf{H}_{\mathbf{m}}^{-1})}\left[\frac{f(\boldsymbol{\delta})}{\gamma}\right] \tag{30}$$

$$= \tilde{C}_d(\mathbf{m}, \gamma) + \mathbb{E}_{\mathcal{N}(\boldsymbol{\delta}'|\mathbf{H}_{\mathbf{m}}^{-1})}\left[\frac{f(\gamma^{1/2}\boldsymbol{\delta}')}{\gamma}\right] \tag{31}$$

$$= \tilde{C}_d(\mathbf{m}, \gamma) + \mathbb{E}_{\mathcal{N}(\boldsymbol{\delta}'|\mathbf{H}_{\mathbf{m}}^{-1})}\left[\frac{O(\|\gamma^{1/2}\boldsymbol{\delta}'\|^3)}{\gamma}\right] \tag{32}$$

$$= \tilde{C}_d(\mathbf{m}, \gamma) + O(\gamma^{1/2}), \tag{33}$$

where $\boldsymbol{\delta}' := \gamma^{-1/2}\boldsymbol{\delta}$. Eq. (20) follows from this inequality. □

## B    DETAILS OF RELATED WORKS

Our method is based on the literature of VAEs, and we aim to improve their model settings for naturally observed high-dimensional data, using the vision domain as an example.

### B.1    COMPARED METHODS

We performed experimental comparisons with several existing related works in Section 6. We first compared our methods with the tuning methods of a weight $\beta$ or a dispersion $\gamma$ in VAEs, and then performed our method with existing VAE-based generative models. The definitions of the reconstruction loss $\mathcal{L}_{\text{rec}}$ and the regularization loss $\mathcal{L}_{\text{reg}}$ follow those of the original VAE as introduced in Section 2.

#### B.1.1    WEIGHT/DISPERSION TUNING METHODS

$\beta$-**VAE (Higgins et al., 2017a).**    Higgins et al. (2017a) have assigned a weight coefficient $\beta$ to the regularization term of the ELBO objective to learn a factorized latent variable. The factorization of the latent variable implies that each independent entry of the latent variable contains a single generative factor of variation, where humans can interpret a latent representation by reading the value of each entry. As the $\beta$-VAE objective given in Eq. (8), the weight parameter $\beta$ steers the balance of the reconstruction and regularization in VAE models. High $\beta$ encourages the element-wise independence of the latent variable $\mathbf{z}$, and low $\beta$ indicates precise reconstruction. Along with the experimental evaluations in (Higgins et al., 2017a), the performance of generative modeling or learning representations is sensitive to $\beta$, prompting several studies to calibrate $\beta$ through training rather than hyperparameter tuning. In addition, from the viewpoint of the information bottleneck method (Tishby et al., 1999; Alemi et al., 2018), it rather means the rate-distortion tradeoff in which higher $\beta$ encourages the latent variable $\mathbf{z}$ to forget the information on the data variable $\mathbf{x}$.

**GECO (Rezende & Viola, 2018).** GECO is a prominent approach to the intuitive tuning of the weight $\beta$ based on a reconstruction error constraint $\mathcal{C}(\mathbf{x}, \hat{\mathbf{x}}_{\boldsymbol{\theta}}(\mathbf{z}))$. The objective of GECO is the Lagrangian $\mathcal{L}_{\boldsymbol{\lambda}}$ containing the sum of the constraint and the regularization term, given as follows:

$$\mathcal{L}_{\boldsymbol{\lambda}} = \mathcal{L}_{\text{reg}} + \boldsymbol{\lambda}^{\mathsf{T}}\mathbb{E}_{p_{\mathcal{D}}(\mathbf{x})q_{\boldsymbol{\phi}}(\mathbf{z}|\mathbf{x})}\left[\mathcal{C}(\mathbf{x}, \hat{\mathbf{x}}_{\boldsymbol{\theta}}(\mathbf{z}))\right], \tag{34}$$

where $\boldsymbol{\lambda}$ denotes a vector of the Lagrange multipliers that has an identical shape with the constraint $\mathcal{C}(\mathbf{x}, \hat{\mathbf{x}}_{\boldsymbol{\theta}}(\mathbf{z}))$. If it is a scalar-valued constraint case, the Lagrangian $\boldsymbol{\lambda}$ is equivalent to the weight $\boldsymbol{\lambda} = \frac{1}{\beta}$ in $\beta$-VAE. The constraint is a reconstruction loss with a hyperparameter threshold $\kappa$, typically given as follows:

$$\mathcal{C}(\mathbf{x}, \hat{\mathbf{x}}_{\boldsymbol{\theta}}(\mathbf{z})) = \|\mathbf{x} - \hat{\mathbf{x}}_{\boldsymbol{\theta}}(\mathbf{z})\|_2^2 - \kappa^2. \tag{35}$$

The GECO algorithm optimizes this constraint in a min-max scheme, where the autoencoding parameters $\boldsymbol{\theta}$ and $\boldsymbol{\phi}$ minimize the objective $\mathcal{L}_{\boldsymbol{\lambda}}$, and the Lagrange multipliers $\boldsymbol{\lambda}$ maximize $\mathcal{L}_{\boldsymbol{\lambda}}$. The regularization is thus maximized, whereas the reconstruction loss is maintained approximately to $\kappa$. Although the value of $\beta$ is difficult to interpret by observing the properties of the latent space, the GECO method enables tuning the reconstruction–regularization balance based on the threshold hyperparameter $\kappa$ of the reconstruction loss.

**Gaussian decoder VAEs with variance $\gamma$.** Several studies (Lucas et al., 2019; Lin et al., 2019; Rybkin et al., 2021) have highlighted the correspondence between a decoder variance parameter $\gamma$ in Gaussian decoders and the weight hyperparameter $\beta$ in $\beta$-VAE. A Gaussian decoder with the global variance $\gamma$ is introduced as follows:

$$p_{\boldsymbol{\theta}}(\mathbf{x}|\mathbf{z}) = \mathcal{N}(\mathbf{x}|\hat{\mathbf{x}}_{\boldsymbol{\theta}}(\mathbf{z}), \gamma\mathbf{I}_L), \tag{36}$$

The reconstruction loss induced by this setting is equivalent to the weight parameter $\beta$ with regard to the autoencoding parameters $(\boldsymbol{\theta}, \boldsymbol{\phi})$, as confirmed below:

$$\nabla_{\boldsymbol{\theta},\boldsymbol{\phi}}[\mathcal{L}_{\text{rec}} + \mathcal{L}_{\text{reg}}] = \nabla_{\boldsymbol{\theta},\boldsymbol{\phi}}\left(\mathbb{E}_{q_{\boldsymbol{\phi}}(\mathbf{x},\mathbf{z})}\left[\log\mathcal{N}(\mathbf{x}|\hat{\mathbf{x}}_{\boldsymbol{\theta}}(\mathbf{z}), \gamma\mathbf{I}_L)\right] + \mathcal{L}_{\text{reg}}\right) \tag{37}$$

$$= \nabla_{\boldsymbol{\theta},\boldsymbol{\phi}}\left(\mathbb{E}_{q_{\boldsymbol{\phi}}(\mathbf{x},\mathbf{z})}\left[\frac{1}{2\gamma}\|\mathbf{x} - \hat{\mathbf{x}}_{\boldsymbol{\theta}}(\mathbf{z})\|_2^2\right] + \mathcal{L}_{\text{reg}}\right) \tag{38}$$

$$= \nabla_{\boldsymbol{\theta},\boldsymbol{\phi}}\left(\frac{1}{\gamma}\mathbb{E}_{q_{\boldsymbol{\phi}}(\mathbf{x},\mathbf{z})}\left[\frac{1}{2}\|\mathbf{x} - \hat{\mathbf{x}}_{\boldsymbol{\theta}}(\mathbf{z})\|_2^2\right] + \mathcal{L}_{\text{reg}}\right) \tag{39}$$

$$= \nabla_{\boldsymbol{\theta},\boldsymbol{\phi}}\left(\frac{1}{\gamma}\mathbb{E}_{q_{\boldsymbol{\phi}}(\mathbf{x},\mathbf{z})}\left[\log\mathcal{N}(\mathbf{x}|\hat{\mathbf{x}}_{\boldsymbol{\theta}}(\mathbf{z}), \mathbf{I}_L)\right] + \mathcal{L}_{\text{reg}}\right) \tag{40}$$

$$= \nabla_{\boldsymbol{\theta},\boldsymbol{\phi}}\frac{1}{\gamma}\cdot\left[\mathcal{L}_{\beta}\Big|_{\beta=\gamma}\right], \tag{41}$$

where $\mathcal{L}_{\beta}$ denotes the $\beta$-VAE objective in Eq. (8). This approach enables the probabilistic interpretation of $\beta$-VAE, where the weight $\beta$ or the variance parameter $\gamma$ represents the degree of the error in estimating the decoding process $p_{\boldsymbol{\theta}}(\mathbf{x}|\mathbf{z})$.

**$\sigma$-VAE (Rybkin et al., 2021).** Rybkin et al. (2021) have proposed a calibration method for the global variance parameter $\gamma$ in VAEs with Gaussian decoders. They have obtained the optimal variance value using the partial derivative of the ELBO objective with regard to the global variance $\gamma$ as follows:

$$\nabla_{\gamma}\mathcal{L} = \nabla_{\gamma}\mathcal{L}_{\text{rec}} \tag{42}$$

$$= \nabla_{\gamma}\left(\frac{1}{\gamma}\mathbb{E}_{q_{\boldsymbol{\phi}}(\mathbf{x},\mathbf{z})}\left[\frac{1}{2}\|\mathbf{x} - \hat{\mathbf{x}}_{\boldsymbol{\theta}}(\mathbf{z})\|_2^2\right] + \frac{M}{2}\log 2\pi\gamma\right) \tag{43}$$

$$= -\frac{1}{\gamma^2}\mathbb{E}_{q_{\boldsymbol{\phi}}(\mathbf{x},\mathbf{z})}\left[\frac{1}{2}\|\mathbf{x} - \hat{\mathbf{x}}_{\boldsymbol{\theta}}(\mathbf{z})\|_2^2\right] + \frac{M}{2\gamma} \tag{44}$$

The equation $\nabla_{\gamma}\mathcal{L} = 0$ produces the optimal variance $\gamma^*$, given as

$$\gamma^* = \frac{2}{M}\mathbb{E}_{q_{\boldsymbol{\phi}}(\mathbf{x},\mathbf{z})}\left[\frac{1}{2}\|\mathbf{x} - \hat{\mathbf{x}}_{\boldsymbol{\theta}}(\mathbf{z})\|_2^2\right]. \tag{45}$$

$\sigma$-VAE uses this optimal value $\gamma^*$ and favorably compares with VAEs with other variance settings including trainable variance and per-pixel variance. The hyperparameter settings experiments in this paper, other than those described in Section 6.1, follow the settings provided by the authors. Specifically, the settings for MNIST and SVHN are: batch size: 128, epoch: 10, learning rate: $10^{-3}$. However, 30,000 steps for CIFAR-10 and Food-101, 40,000 steps for CelebA.

### B.1.2 VARIATIONAL AUTOENCODING GENERATIVE MODELS

**DFCVAE (Hou et al., 2019).** Hou et al. (2019) have proposed DFCVAE, a VAE-based model utilizing the hidden features of pretrained image classification models instead of the per-pixel MSE loss. This method replaces the reconstruction loss with the perceptual loss as follows:

$$\mathcal{L}_{\text{rec}} = \frac{1}{2N_{\text{layer}}} \sum_{l=1}^{N_{\text{layer}}} \frac{1}{C^{(l)}H^{(l)}W^{(l)}} \sum_{c=1}^{C^{(l)}} \sum_{i=1}^{H^{(l)}} \sum_{j=1}^{W^{(l)}} \left[ x_{c,h,w}^{(l)} - \hat{x}_{c,h,w}^{(l)}(\mathbf{z}) \right]^2, \tag{46}$$

where $x_{c,h,w}^{(l)}$ and $\hat{x}_{c,h,w}^{(l)}(\mathbf{z})$ denote the hidden features of $\mathbf{x}$ and $\hat{\mathbf{x}}_{\boldsymbol{\theta}}(\mathbf{z})$ at the $(h, w)$-th entry of the $l$-th layer, respectively. The hidden features consist of $N_{\text{layer}}$ layers, and the $l$-th layer contains $H^{(l)} \times W^{(l)}$-sized feature maps with $C^{(l)}$ channels ($l = 1, 2, \ldots, N_{\text{layer}}$). Although the original authors have reported better generative results in the vision domain, the balance between the reconstruction and regularization terms remains an issue because the perceptual loss in Eq. (46) simply takes the mean of hidden features with the degree of freedom in its scale. In this paper, the hyperparameter settings for experiments, except those described in Section 6.1, were based on the settings provided by the authors. Specifically, the following parameters were used for all datasets: $\beta = 1$, batch size: 128, number of epochs: 100, learning rate: $10^{-3}$ (exponentially decayed to $10^{-4}$ during the training process of epochs).

**VAE-GAN (Larsen et al., 2016).** Larsen et al. (2016) have proposed VAE-GAN, a hybrid model of VAEs and generative adversarial networks (GANs). The VAE-GAN model introduces a discriminator for a metric of the reconstruction term, and the VAE-GAN objective comprises three losses $\mathcal{L}_{\text{Enc}}$, $\mathcal{L}_{\text{Dec}}$, and $\mathcal{L}_{\text{Dis}}$ respectively for the encoder, decoder, and discriminator networks. These three losses are given as follows:

$$\mathcal{L}_{\text{Enc}} = \mathcal{L}_{\text{reg}} + \mathcal{L}_{\text{rec}}, \tag{47}$$

$$\mathcal{L}_{\text{Dec}} = \gamma \mathcal{L}_{\text{rec}} - \mathcal{L}_{\text{GAN}}, \tag{48}$$

$$\mathcal{L}_{\text{Dis}} = \mathcal{L}_{\text{GAN}}, \tag{49}$$

where $\mathcal{L}_{\text{GAN}}$ denotes an adversarial loss using the discriminator $\text{Dis}(\cdot)$, given as

$$\mathcal{L}_{\text{GAN}} = \mathbb{E}_{p_{\mathcal{D}}(\mathbf{x})} \left[ \log \text{Dis}(\mathbf{x}) \right] + \mathbb{E}_{q_{\boldsymbol{\phi}}(\mathbf{z})} \left[ \log \text{Dis}(\hat{\mathbf{x}}_{\boldsymbol{\theta}}(\mathbf{z})) \right] + \mathbb{E}_{\pi(\mathbf{z})} \left[ \log(1 - \text{Dis}(\hat{\mathbf{x}}_{\boldsymbol{\theta}}(\mathbf{z}))) \right]. \tag{50}$$

In VAE-GAN models, the reconstruction loss $\mathcal{L}_{\text{rec}}$ contains the MSE of hidden features in the discriminator as well as that of the data variable. In this paper, the hyperparameter settings for experiments, except those described in Section 6.1, were based on the settings provided by the authors. Specifically, the following parameters were used for all datasets: $\beta = 1$, $\gamma = 1$, batch size: 128, number of epochs: 100, learning rate: $10^{-3}$ (exponentially decayed to $10^{-4}$ during the training process of epochs).

**2-Stage VAE (Dai & Wipf, 2019).** Dai & Wipf (2019) have diagnosed VAE models theoretically and proposed 2-Stage VAE, an improved VAE framework based on two stages of the training process. In the first stage, a preliminary representation is learned in a Gaussian VAE. The aggregated posterior of this preliminary representation does not always match the prior, as $q_{\boldsymbol{\phi}}(\mathbf{z}) \not\approx \pi(\mathbf{z})$ in the standard VAE. To address this issue, a second VAE is trained using the samples of the preliminary representation as an observation variable. Thus, the latent representation is learned in the latent variable of the second VAE. The training process consisting of these two stages improves the generative quality of VAEs by matching the aggregated posterior $q_{\boldsymbol{\phi}}(\mathbf{z})$ and the prior $\pi(\mathbf{z})$ in the second stage of training. In this paper, the hyperparameter settings for experiments, except those described in Section 6.1, were based on the settings provided by the authors. Specifically, the following parameters were used for all datasets: batch size: 128, number of epochs: 100, learning rate: 3e-4.

**Soft-IntroVAE (Daniel & Tamar, 2021).** Daniel & Tamar (2021) have proposed Soft-IntroVAE, a VAE-based adversarial generative model using its encoder as a discriminator. Although the main idea of adversarial VAEs using their encoder had been proposed in (Huang et al., 2018), the training with its hinge loss is practically difficult to stabilize. To solve this problem, Soft-IntroVAE maximizes smooth exponential objectives, where an encoder objective $\mathcal{L}_{\mathrm{Enc}}$ and a decoder objective $\mathcal{L}_{\mathrm{Dec}}$ are maximized adversarially as follows:

$$\mathcal{L}_{\mathrm{Enc}} = \mathcal{L}_{\mathrm{ELBO}}(\mathbf{x}) - \frac{1}{\alpha}\exp(\alpha\mathcal{L}_{\mathrm{ELBO}}(\hat{\mathbf{x}}_{\boldsymbol{\theta}}(\mathbf{z}))), \tag{51}$$

$$\mathcal{L}_{\mathrm{Dec}} = \mathcal{L}_{\mathrm{ELBO}}(\mathbf{x}) + \gamma\mathcal{L}_{\mathrm{ELBO}}(\hat{\mathbf{x}}_{\boldsymbol{\theta}}(\mathbf{z})), \tag{52}$$

where $(\mathbf{x}, \mathbf{z}) \sim q_{\boldsymbol{\phi}}(\mathbf{x}, \mathbf{z})$, and $\alpha = 2$ and $\gamma = 1$ in the original authors' settings (Daniel & Tamar, 2021) In this paper, the hyperparameter settings for experiments, except those described in Section 6.1, were based on the settings provided by the authors. Specifically, the following parameters were used for all datasets: $\beta_{kl} = 1$, $\beta_{rec} = 1$, $\gamma_r = 10^{-8}$, batch size: 128, number of epochs: 100, learning rate: 2e-4.

### B.2 Dissimilarity Functions for Visual Domain

In this subsection, we introduce the dissimilarities $d$ mentioned in Section 6, adopted for the vision domain as investigated in the image quality assessment (IQA) field. Deviating from the original definition, several dissimilarities are multiplied by $-1$ and constants are added to satisfy the non-negativity in Eq. (12). Let $\mathbf{x}$ and $\mathbf{m}$ denote the variable under comparison, and $M$ the number of their elements. In the EDDF density, $\mathbf{x}$ and $\mathbf{m}$ correspond to the reference and distorted images, respectively.

#### B.2.1 Classical Dissimilarities

**Mean squared error (MSE).** MSE is a commonly adopted loss function of general variables, given as follows:

$$d(\mathbf{x}, \mathbf{m}) = \frac{1}{M}\|\mathbf{x} - \mathbf{m}\|_2^2. \tag{53}$$

This loss is equivalent to the squared $L_2$ norm up to the coefficient $\frac{1}{M}$.

**Mean absolute error (MAE).** MAE is another well-known loss function of general variables, given as follows:

$$d(\mathbf{x}, \mathbf{m}) = \frac{1}{M}\|\mathbf{x} - \mathbf{m}\|_1. \tag{54}$$

As in the MSE loss, this is equivalent to the $L_1$ norm up to $\frac{1}{M}$.

**Cross entropy (CE).** The cross entropy loss is an oft-used loss function for binary variables, given as follows:

$$d(\mathbf{x}, \mathbf{m}) = -\sum_{i=1}^{M} x_i \log m_i - \sum_{i=1}^{M}(1 - x_i)\log(1 - m_i), \tag{55}$$

where $x_i$ and $m_i$ denote the $i$-th entry ($i = 1, 2, \ldots, M$) of $\mathbf{x}$ and $\mathbf{m}$, respectively.

**Cosine Similarity.** The cosine similarity is also a well-known similarity function that compares the direction of variables, given as follows:

$$d(\mathbf{x}, \mathbf{m}) = 1 - \frac{\mathbf{x}^{\mathsf{T}}\mathbf{m}}{\|\mathbf{x}\|_2\|\mathbf{m}\|_2}. \tag{56}$$

#### B.2.2 Dissimilarity Indices Based on Hand-Crafted Features

**Structural similarity (SSIM) (Wang et al., 2004).** SSIM is a similarity function for assessing the image quality, transformed into a dissimilarity function as follows:

$$d(\mathbf{x}, \mathbf{m}) = 1 - \frac{(2\mu_{\mathbf{x}}\mu_{\mathbf{m}} + c_1)(2\sigma_{\mathbf{x},\mathbf{m}} + c_2)}{(\mu_{\mathbf{x}}^2 + \mu_{\mathbf{m}}^2 + c_1)(\sigma_{\mathbf{x}}^2 + \sigma_{\mathbf{m}}^2 + c_2)}, \tag{57}$$

where $\mu.$ and $\sigma.$ respectively denote the mean and (co)variance in the window of the image denoted by the subscript, $c_1 = (0.01L)^2$, $c_2 = (0.03L)^2$, and $L$ denotes the dynamic range of the images. Because SSIM is a bounded function with the range $[0, 1]$, we used it as a dissimilarity function $1 - \text{SSIM}$. SSIM has many derived methods for measuring similarity in the image quality assessment (IQA) task (Wang & Li, 2011; Sheikh & Bovik, 2006; Zhang et al., 2011; Zhang & Li, 2012; Xue et al., 2014; Zhang et al., 2014; Balanov et al., 2015). We present these methods to provide the comparison details of Table 4.

**IW-SSIM (Wang & Li, 2011).** IW-SSIM is an extended similarity function based on SSIM (Wang et al., 2004), introducing the idea of information content-weighted pooling. The computation of IW-SSIM is based on the mutual information between input signals and their perception channels.

**VIFp (Sheikh & Bovik, 2006).** VIFp is a similarity function proposed in Sheikh & Bovik (2006), based on the ratio of distorted and reference image information. These two types of image information are computed by the mutual information between the input and output of the human visual system (HVS) channel. The HVS channel is based on the visual perception of humans, enabling the measurement of image fidelity based on information that humans can perceive.

**FSIM (Zhang et al., 2011).** FSIM is an image similarity index based on low-level image features with which the HVS understands images. The FSIM function consists of two parts: phase congruency and image gradient magnitude. These two parts complementarily function as the features of FSIM, where the phase congruency part primarily measures the significance of local structures, and the image gradient magnitude part secondarily captures contrast information.

**SR-SIM (Zhang & Li, 2012).** SR-SIM is a visual similarity function based on spectral residual visual saliency. This similarity function assumes that visual saliency maps have a close relation to the quality of human perception, and the similarity is a weighted sum of local similarities by the spectral residual visual saliency map proposed by Hou & Zhang (2006).

**GMSD (Xue et al., 2014).** GMSD is an IQA index based on the global variation of spacial gradient-based local quality maps, considering the sensitivity of image gradients to visual distortion. It measures the per-pixel similarity of image gradient magnitudes and uses a pooling method to estimate the standard deviation of the gradient magnitude similarity. The original paper (Xue et al., 2014) reported that these strategies yield a fast running time compared with other similarity indices, including SSIM (Wang et al., 2004), IW-SSIM (Wang & Li, 2011), and VIFp (Sheikh & Bovik, 2006).

**VSI (Zhang et al., 2014).** VSI is a visual similarity index based on visual saliency, a crucial part of the HVS for the IQA field. This index uses visual saliency in its computation, which plays a key role in the prediction performance in measuring image quality.

**DSS (Balanov et al., 2015).** DSS is an image similarity index based on the subbands of the discrete cosine transform (DCT). The DSS index measures the structural changes in the DCT domain, where signals of different frequencies can be separately compared. Although the computation cost of DSS is higher than those of simpler similarity indices, such as PSNR and SSIM, the DSS achieves better performance in the IQA task (Balanov et al., 2015).

**HaarPSI (Reisenhofer et al., 2018).** HaarPSI is an image similarity measure established upon the Haar wavelet-based perceptual similarity. HaarPSI is computationally inexpensive because the Haar wavelets can be efficiently computed (Stanković & Falkowski, 2003), and the original paper (Reisenhofer et al., 2018) has reported that HaarPSI had achieved a better correlation with human quality assessment and moderately fast computation than other image quality indices.

### B.2.3 DEEP LEARNING (DL)-BASED DISSIMILARITIES

**Content and Style Scores (Gatys et al., 2016).** Content Score and Style Score have been proposed by Gatys et al. (2016) for the style transfer task using pretrained convolutional neural networks (CNNs). To control the artistic style of images, Gatys et al. (2016) separate and recombine the content

Table 6: Model architecture of encoders for the neural networks (NNs) $\boldsymbol{\mu}_\phi$ and $\log \boldsymbol{\sigma}_\phi$. Conv means a convolution layer, and FC denotes a fully-connected layer. We applied the GELU activation (Hendrycks & Gimpel, 2016) to the output of each layer, except the last layer, and used batch normalization after each convolution layer.

| Layer | Input | Output | Settings |
|-------|-------|--------|----------|
| \multicolumn{4}{c}{Inverse sigmoid function $\sigma^{-1}$} | | | |
| Conv | $3 \times 64 \times 64$ | $32 \times 32 \times 32$ | kernel $4 \times 4$, stride 2, padding 1 |
| Conv | $32 \times 32 \times 32$ | $64 \times 16 \times 16$ | kernel $4 \times 4$, stride 2, padding 1 |
| Conv | $64 \times 16 \times 16$ | $128 \times 8 \times 8$ | kernel $4 \times 4$, stride 2, padding 1 |
| Conv | $128 \times 8 \times 8$ | $256 \times 4 \times 4$ | kernel $4 \times 4$, stride 2, padding 1 |
| FC | 4096 | 256 | |
| FC | 256 | $L$ | for each of $\boldsymbol{\mu}_\phi$ and $\log \boldsymbol{\sigma}_\phi$ |

and style information of images, based on the hidden features of pretrained CNNs. In this method, they use separate losses to match content and style, where such losses can also be used for the image assessment from different perspectives. These losses are constructed based on the CNNs, which have tractable gradients and smoothness for applying stochastic gradient descent.

**LPIPS (Zhang et al., 2018).** LPIPS is a DL-based image similarity metric that uses the features of pretrained CNNs as a perceptual metric space. LPIPS measures the perceptual distance between images using the weighted mean of hidden CNN features. Hidden features are provided by pretrained image classification networks, such as the VGG networks (Simonyan & Zisserman, 2015). The original paper (Zhang et al., 2018) has reported that LPIPS outperforms traditional shallow similarities such as SSIM in perceptual tasks and the IQA.

**PieApp (Prashnani et al., 2018).** PieAPP is an image-error assessment index based on learning pair-wise preference. The pair-wise preference is annotated by humans and contains the preference of humans between images distorted in different ways, which becomes the target variable of learning PieAPP (Prashnani et al., 2018). The original paper (Prashnani et al., 2018) has reported its effectiveness in the IQA task over traditional similarities including SSIM and several DL-based methods such as LPIPS (Zhang et al., 2018).

**DISTS (Ding et al., 2022).** DISTS is an IQA similarity based on structure and texture, that uses injective and differentiable CNNs for measuring distortion with beneficial gradients in their feature maps. It has been reported by Ding et al. (2022) that the IQA performance of DISTS compares favorably with those of other similarity indices such as LPIPS. The original paper of DISTS (Ding et al., 2022) has also reported the effectiveness of DISTS in texture classification tasks and its insensitivity to mild geometric transformations.

## C  ARCHITECTURE DETAILS

We used the encoder architecture shown in Table 6 and the decoder architecture in Table 7 in all models experimented. These architectures are simple and lightweight networks consisting of the fully-connected, convolution, and transposed convolution layers. We trained each of these variational autoencoder (VAE) networks using the Adam (Kingma & Ba, 2015) optimizer with a learning rate of $10^{-3}$, exponentially decayed into $10^{-4}$ during the training process of 100 epochs. A single run of VAEs with this architecture took approximately an hour in a single GPU of GeForce® RTX 2080 Ti™, which is considerably less computationally expensive than adversarial approaches.

Table 7: Model architecture of decoders for the NN $\hat{\mathbf{x}}_{\boldsymbol{\theta}}(\mathbf{z})$. TrConv means a transposed convolution layer, and FC denotes a fully-connected layer. We applied the GELU activation (Hendrycks & Gimpel, 2016) to the output of each layer, excluding the last layer, and used batch normalization after each transposed convolution layer, except for the last one.

| Layer | Input | Output | Settings |
|---|---|---|---|
| FC | $L$ | 256 | |
| FC | 256 | 4096 | |
| TrConv | $256 \times 4 \times 4$ | $128 \times 8 \times 8$ | kernel $4 \times 4$, stride 2, padding 1 |
| TrConv | $128 \times 8 \times 8$ | $64 \times 16 \times 16$ | kernel $4 \times 4$, stride 2, padding 1 |
| TrConv | $64 \times 16 \times 16$ | $32 \times 32 \times 32$ | kernel $4 \times 4$, stride 2, padding 1 |
| TrConv | $32 \times 32 \times 32$ | $3 \times 64 \times 64$ | kernel $4 \times 4$, stride 2, padding 1 |
| | | Sigmoid activation $\sigma$ | |

## D  DATASET DETAILS

As shown in Table 2, we used several vision datasets including various backgrounds, where all images were resized to $64 \times 64$ pixels for experiments for a fair comparison between datasets. For the experiments, we used the `torchvision`[2] implementation for the five datasets presented below.

**MNIST (LeCun et al., 1998).**   The MNIST dataset consists of 70,000 binary images of hand-written numerical digits with class labels ranging from 0 to 9. The original author provides 60,000 of these images as the training set and 10,000 as the test set separately. We further divided the training set into 54,000 for training and 6,000 for validation randomly. The original dataset is available online.[3]

**SVHN (Netzer et al., 2011).**   The SVHN dataset comprises color digit images obtained from house numbers, with the training set containing contains 73,257 images and the test set comprising 26,032 images. Each of these images has a class label that corresponds to the digit located in the center of the image, whereas some of the images contain peripheral digits other than the center digit. The original author provides this dataset online.[4]

**CelebA (Liu et al., 2015).**   CelebA is a color image dataset containing 202,599 celebrity face images with 40 binary attributes for each. The original author provides a data split online,[5] which includes 162,770 images for the training set, 19,867 images for the validation set, and 19,962 images for the test set. The original author provides this dataset online.[6] Although the original resolution is $178 \times 218$ pixels, we further cropped the center $144 \times 144$ pixels of each image for the experiments.

**Food-101 (Bossard et al., 2014).**   The Food-101 dataset comprises color food images of 101 categories, totaling 101,000 images. We randomly split them into 68,175 images for the training set, 7,575 images for the validation set, and 25,250 images for the test set. Although this dataset may contain images with inaccurate category labels as the original author mentioned (Bossard et al., 2014), such noise does not affect the experiments in this paper because we evaluated the unsupervised models. The original dataset is available online.[7]

**CIFAR-10 (Krizhevsky & Hinton, 2009).**   The CIFAR-10 dataset consists of 60,000 color images with 10 categories of vehicles and animals. Although this dataset originally contains 50,000 training images and 10,000 test images, we randomly select 5,000 images from the original training set to create a validation set. CIFAR-10 is a more complex dataset than the aforementioned datasets because

---

[2]https://github.com/pytorch/vision
[3]http://yann.lecun.com/exdb/mnist/
[4]http://ufldl.stanford.edu/housenumbers/
[5]https://mmlab.ie.cuhk.edu.hk/projects/CelebA.html
[6]https://mmlab.ie.cuhk.edu.hk/projects/CelebA.html
[7]https://data.vision.ee.ethz.ch/cvl/datasets_extra/food-101/

it contains a wide range of categories, and hence, its images are not positionally aligned. The original dataset is available online.[8]

## E    EVALUATION INDEX DETAILS

Using the Inception-v3 (Szegedy et al., 2016) features trained in ImageNet (Deng et al., 2009), Fréchet Inception Distance (FID) is defined as the 2-Wasserstein distance of real and generated image features, as follows:

$$
\begin{aligned}
\text{FID} = &\|\boldsymbol{\mu}_{\text{gen}} - \boldsymbol{\mu}_{\text{real}}\|_2^2 \\
&+ \text{tr}\left(\boldsymbol{\Sigma}_{\text{gen}} + \boldsymbol{\Sigma}_{\text{real}} - 2\boldsymbol{\Sigma}_{\text{gen}}^{1/2}\boldsymbol{\Sigma}_{\text{real}}\boldsymbol{\Sigma}_{\text{gen}}^{1/2}\right),
\end{aligned}
\tag{58}
$$

where $(\boldsymbol{\mu}_{\text{gen}}, \boldsymbol{\Sigma}_{\text{gen}})$ denote the Inception-v3 feature mean and covariance of the generated images, and $(\boldsymbol{\mu}_{\text{real}}, \boldsymbol{\Sigma}_{\text{real}})$ represent those of real images. Kernel Inception Distance (KID) is another approach for evaluating the generative performance using the squared maximum mean discrepancy (MMD) estimator with a third-order polynomial kernel function $k$ instead of the 2-Wasserstein distance, as follows:

$$
\begin{aligned}
\text{KID} = &\mathbb{E}_{p_{\text{gen}}(f)}\mathbb{E}_{p_{\text{gen}}(f')}\left[k(f, f')\right] \\
&+ \mathbb{E}_{p_{\text{real}}(f)}\mathbb{E}_{p_{\text{real}}(f')}\left[k(f, f')\right] \\
&- 2\mathbb{E}_{p_{\text{gen}}(f)}\mathbb{E}_{p_{\text{real}}(f')}\left[k(f, f')\right],
\end{aligned}
\tag{59}
$$

where $p_{\text{gen}}$ and $p_{\text{real}}$ denote the Inception-v3 feature distributions of the generated and real images, respectively.

## F    QUALITATIVE RESULTS

In addition to the quantitative evaluations in Tables 3 and 5, we compared the proposed method with existing variational autoencoding models on reconstruction and generation in Figs. 3 and 4, respectively. In Fig. 3, VAEs with exponential dissimilarity-dispersion family (EDDF) decoders denoted by "Ours" appears to have retained the style and details of the input images, especially on relatively complex datasets such as Food-101 and CIFAR-10. Although DFCVAE (Hou et al., 2019) also reconstructed detailed edges using the pretrained CNN, the reconstruction does not appear to retain global style such as overall color. In Fig. 4, VAEs based on the proposed method successfully generated unseen images. Compared with other VAE-based methods, the generated image samples using the proposed method seems to inhibit over-smoothing frequently observed in VAEs due to stochastic training.

## G    EMPIRICAL VALIDATION OF "WELL-TRAINED AUTOENCODER" ASSUMPTION

To investigate the validity of the "well-trained autoencoder" assumption $\gamma \ll 1$ made in Section 4, we present the curve of the approximated calibrated dispersion $\tilde{\gamma}^*$ in several runs of the proposed method in Fig. 5. Although there was a slight tendency of overfitting, their values diverged to small values on the order of $10^{-2}$. These results suggest that the global dispersion $\gamma$ is small even in high-dimensional natural data, as theoretically shown by Dai & Wipf (2019). Therefore, the "well-trained autoencoder" assumption is suitable for practical cases, at least in the vision domain.

---

[8] https://www.cs.toronto.edu/~kriz/cifar.html

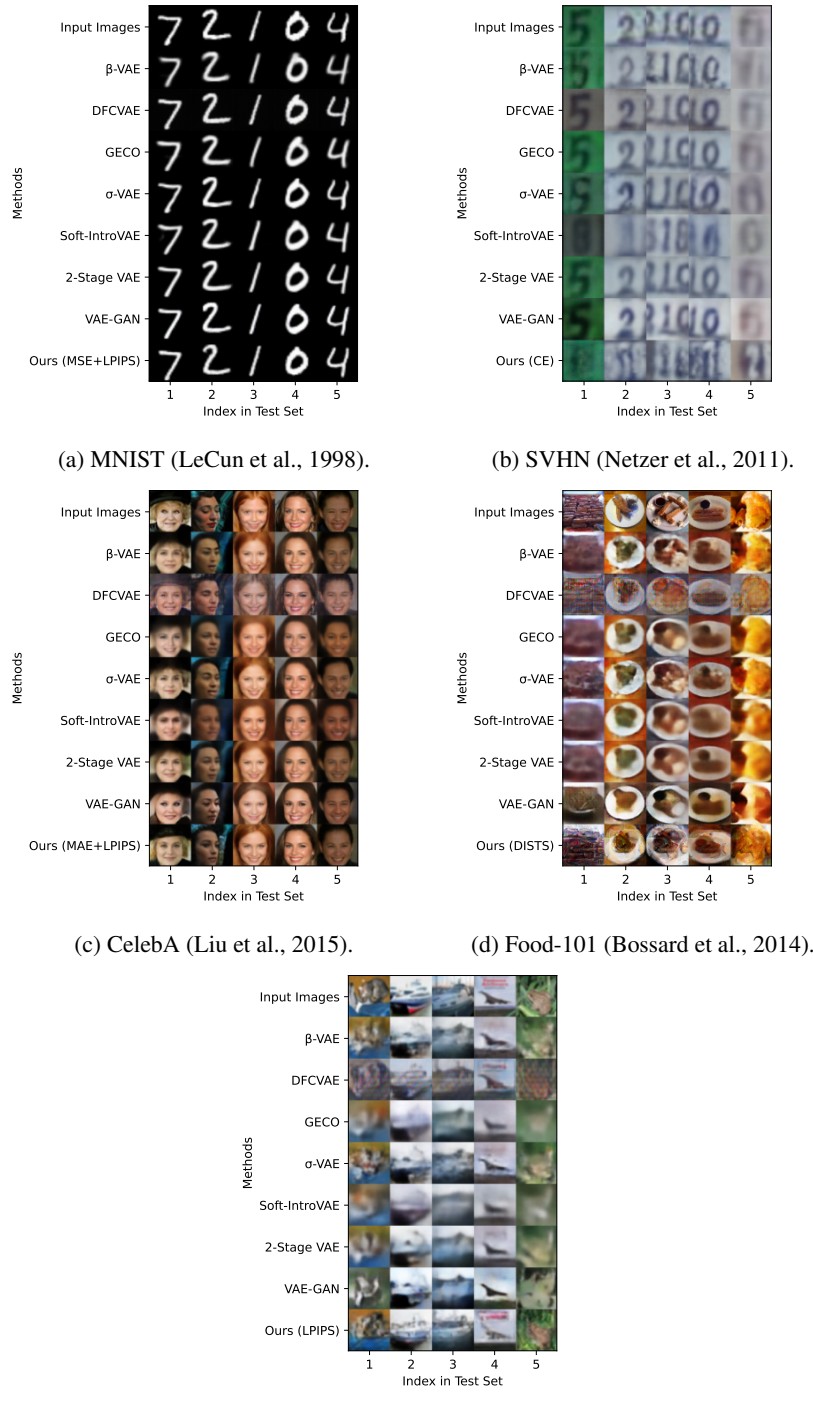

(a) MNIST (LeCun et al., 1998).   (b) SVHN (Netzer et al., 2011).

(c) CelebA (Liu et al., 2015).   (d) Food-101 (Bossard et al., 2014).

(e) CIFAR-10 (Krizhevsky & Hinton, 2009).

Figure 3: Qualitative comparisons in terms of reconstruction. The images on the "Input Images" rows represent the images input into the models shown below these rows. Each column corresponds to a sample of the test set. The words in parentheses appended to "Ours" represent the dissimilarity function selected through FID validation, and "CE" denotes the Cross Entropy dissimilarity.

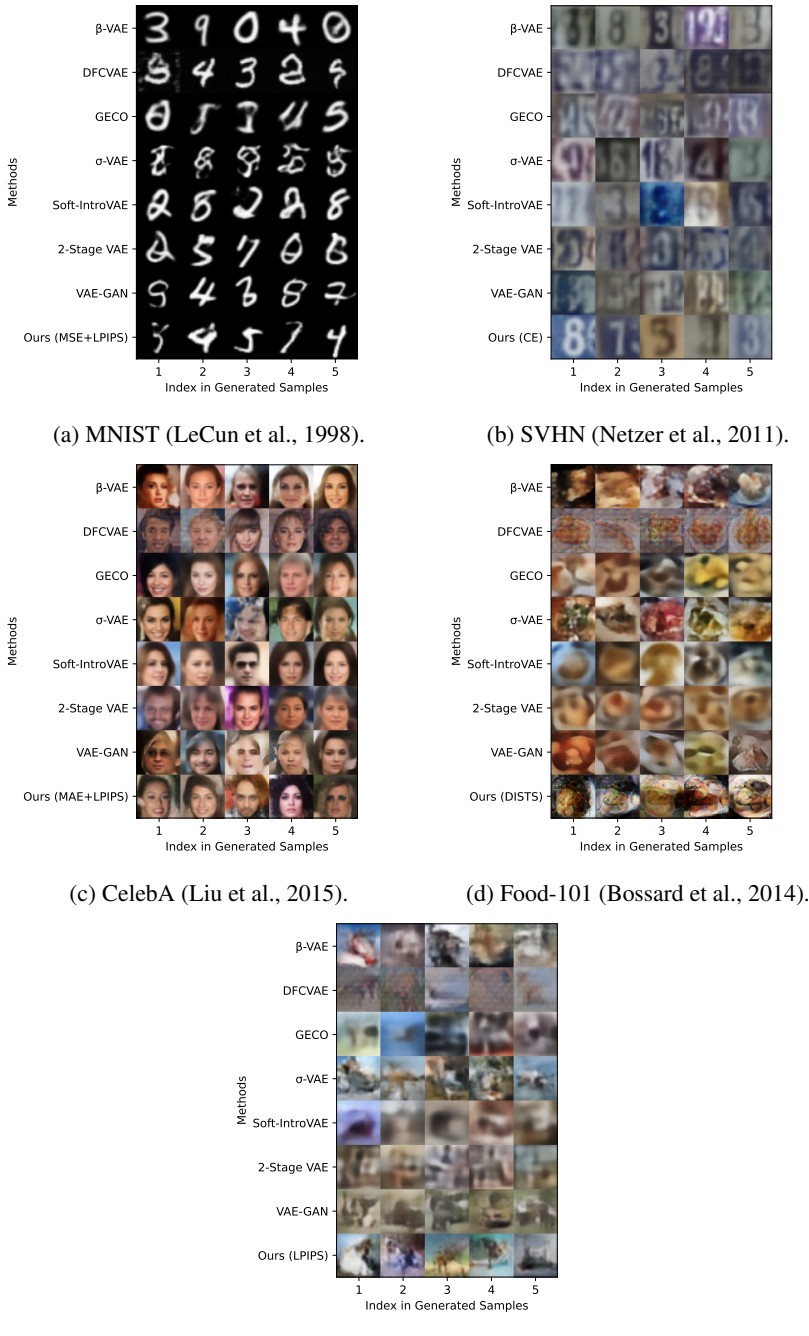

(a) MNIST (LeCun et al., 1998).

(b) SVHN (Netzer et al., 2011).

(c) CelebA (Liu et al., 2015).

(d) Food-101 (Bossard et al., 2014).

(e) CIFAR-10 (Krizhevsky & Hinton, 2009).

Figure 4: Qualitative comparisons in terms of generation. Each image sample is independently drawn from the generative model $p_{\theta}(\mathbf{x})$. To draw samples, we first drew a latent sample from the prior $\pi(\mathbf{z})$ and then obtained an image sample using the decoder (generator) of the model corresponding to the row. Quantitative comparisons are shown in Table 5.

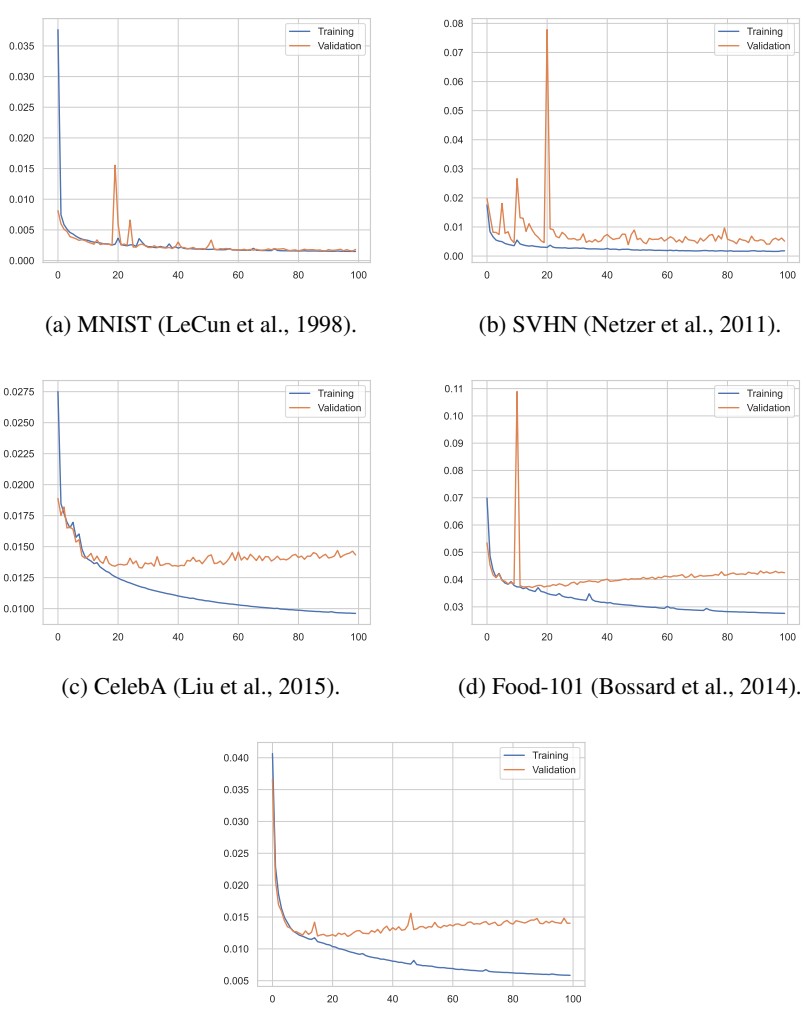

(a) MNIST (LeCun et al., 1998).

(b) SVHN (Netzer et al., 2011).

(c) CelebA (Liu et al., 2015).

(d) Food-101 (Bossard et al., 2014).

(e) CIFAR-10 (Krizhevsky & Hinton, 2009).

Figure 5: Curves of the approximated dispersion $\tilde{\gamma}^*$ implicitly learned via the log-expected dissimilarity loss $\mathcal{L}_{\mathrm{rec}}^{\widetilde{\times}}$. The curves "Training" and "Validation" denote the $\tilde{\gamma}^*$ values of the training and validation sets, respectively.

