# OpenReview forum: "Incorporating Domain Knowledge in VAE Learning via Exponential Dissimilarity-Dispersion Family"
_ICLR.cc/2024/Conference — Submitted to ICLR 2024_

### Official Review · Reviewer_MY4X · 2023-10-21

**Soundness:** 2 fair
**Presentation:** 3 good
**Contribution:** 2 fair
**Rating:** 5
**Confidence:** 4

**Summary:**

This paper introduces the exponential dissimilarity-dispersion family (EDDF) as the distribution of the decoder $p_{\theta}(x|z)$. Adopting EDDF can use different dissimilarity functions to define the reconstruction loss and provide an implicit optimization of the dispersion parameter $\gamma$ that balances the rate-distortion trade-off without hand-tuning as $\beta$-VAE.

**Strengths:**

1. The paper is well-written and easy to follow.
2. The proposed method is simple and well-motivated.
3. Code is released.

**Weaknesses:**

1. The general derivation is quite similar to $\sigma$-VAE, except for the extension of MSE to different dissimilarity functions.
2. Although the experiments are designed from different perspectives to verify the effectiveness of the proposed methods, I have the following concerns:
    * When choosing MSE as the similarity metric, why does there exist a performance difference between $\sigma$-VAE and the proposed method in Tab.3?
    * In Tab.5, different dissimilarities $d$ are chosen based on the validation set for different datasets. How are the hyper-parameters tuned for the baselines? It seems that $\beta$-VAE with different $\beta$ in Tab.3 performs better than some baselines. Why not compare to $\beta$-VAE with a hyper-parameter tuned on the validation set?

**Questions:**

Please see the previous section.

---

> ### Author Response · Authors · 2023-11-15
> **Response to Reviewer MY4X**
>
> Thank you for reviewing the manuscript. We will answer your questions below.
>
> **The general derivation is quite similar to $\sigma$-VAE, except for the extension of MSE to different dissimilarity functions.**
> The σ-VAE proposes an approach to resolve the balancing and tuning issues faced by β-VAE and indeed shares a similar concept with our paper. While the σ-VAE focuses on calibrating β in the decoder, our proposed method significantly differs by using dissimilarity functions to extend the functionality of VAEs themselves, expanding their theoretical framework while maintaining probabilistic consistency. Additionally, our method allows for the automatic adjustment of scale parameters, eliminating the need for manual tuning of the decoder's variance, unlike σ-VAE. Moreover, experiments have confirmed that our proposed method demonstrates equal or superior performance to σ-VAE. We believe these novelties represent a significant advancement in the extensibility of VAEs beyond what σ-VAE offers.
>
> **When choosing MSE as the similarity metric, why does there exist a performance difference between $\sigma$-VAE and the proposed method in Tab.3?**
> This aspect is one of the key points where the advantages of our proposed method are most evident. As stated in Viewpoint (i) of the introduction, the focus of our paper is the necessity of introducing a family of distributions to utilize prior information in VAEs, which already have a theoretical foundation. While previous studies have attempted to introduce additional mechanisms to balance reconstruction and regularization errors (e.g., $\sigma$-VAE, Rybkin et al., 2021), these approaches often deviated from the probabilistic framework, posing a challenge. Thus, our paper aims to resolve these traditional issues by attempting to learn using domain knowledge in the form of dissimilarity functions while maintaining the probabilistic model framework inherent in VAEs. Specifically, we propose a unified approach that balances reconstruction and regularization through variational inference backed by a probabilistic understanding of distribution families, rather than resolving issues through additional mechanisms or regularization.
> The experimental results in Table 3 show that the advantages of $\sigma$-VAE can be replicated in comparison with β-VAE, and these benefits are demonstrated. Furthermore, our proposed method has shown superiority over these techniques, particularly demonstrating the effectiveness of automatic tuning including the dispersion $\gamma$ in complex datasets such as Food-101 and CIFAR-10. This indicates that the proposed method, compared to $\sigma$-VAE, provides a more nuanced capture of data characteristics, which can lead to differences in performance.
>
> **In Tab.5, different dissimilarities ... tuned on the validation set?**
> First, regarding Table 5, the hyperparameters for both the baseline and proposed methods were determined based on the FID from the validation set. The dissimilarity functions selected in our method are detailed in Figure 4.
> As for Table 3, rather than optimizing β or evaluating the highest quality resolution, we focused on validating how different values of β affect performance. This allows us to directly observe the effects of changes in β. We compare these performances to those of the automatically determined γ. We will add to the revised manuscript the points that were insufficiently explained.
>
> If the explanations provided have resolved your concerns, we would appreciate it if you could reconsider the score given.

---

> ### Author Response · Authors · 2023-11-21
> **2 Days Left Reminder: Seeking Reviewer Feedback and Resolving Concerns for Reviewer MY4X**
>
> We appreciate your inquiry and thank you for your advice. We would like to make sure our response meets your needs. If you have any more questions or if there's anything in our response that doesn't fully address your concerns, please let us know. Your feedback is important to us and we'd be happy to discuss further. With just two days left in the discussion period, we're wondering if you've had a chance to review our response and if there's anything else we can help with.

---

> ### Comment · Reviewer_MY4X · 2023-11-21
>
> Thanks for your reply! My concern remains.
>
> * When using MSE as the similarity metric, mathematically, I do not see any evident difference between the proposed method and $\sigma$-VAE (Rybkin et al., 2021). In Rybkin et al., 2021, the authors have mentioned just below eq(6) that $\sigma$ can be selected by minimizing the weighted MSE loss, which is indicated as the optimal $\sigma$ and **also does not require the manual tuning of $\sigma$**.
>
> * W.r.t the experiments:
>     - In Table 3, for SVHN, $\beta=0.1$, you have FID = 15.97, KID=0.0897, the FID is very close to that of the proposed method in Table 5, and the KID is much better. Moreover, the results of $\beta$-VAE in Table 3 are only based on six choices of $\beta$. My point is: why not compare to $\beta$-VAE with **more choices of $\beta$** in Table 5 to illustrate that the proposed method really outperforms?
>     - **How are the hyper-parameters tuned for the baselines?** I want to see how you set the hyper-parameter range for each baseline and what are the specific hyper-parameters for each baseline. **I have already noted what you marked as blue when I was reviewing the paper**.
>      - As you have incorporated the "domain knowledge" (actually weird to say so, because it's not predefined, but tuned as a hyper-parameter) in the form of dissimilarity functions to achieve better performance, why not directly indicate in Table 5 the chosen dissimilarity function? (I guess you mean detailed in Table 4?) I can relate this information through the two tables, but it takes additional time for the readers.

---

> > ### Author Response · Authors · 2023-11-22
> > **Response to Reviewer MY4X (2)**
> >
> > Thank you for your question. I understand your intention, and I will provide a detailed response below.
> >
> > **When using MSE as the similarity ...  the manual tuning of $\sigma$.**
> > Thank you for your feedback. Indeed, from the perspective that MSE is used as a similarity metric, it may appear that manual adjustment of $\gamma$ (equivalent to $\sigma$ in $\sigma$-VAE) is not required, as it seems similar. However, $\sigma$-VAE estimates the optimal $\sigma$ using a Monte Carlo method with mini-batches, while the proposed approach, even in the case of MSE, can ignore it as a constant term when taking the logarithm and does not perform estimation in the first place. This represents a clear algorithmic difference.
> >
> > **In Table 3, for SVHN, ... proposed method really outperforms?**
> > Thank you for your advice. Firstly, the experiments in Table 3 and Table 5 have significantly different purposes. In Table 3, the main objective is to evaluate the adjustment algorithms for $\beta, \gamma, \sigma$. Therefore, experiments are conducted by assuming a comparison with $\sigma$-VAE and a simplification of the problem setting to MSE. Indeed, by comparing the values in Table 3 and Table 5, it can be said that $\beta$-VAE achieves competitive accuracy on simpler datasets like MNIST and SVHN, given the assumption of comparison with $\sigma$-VAE and the simplification to the MSE problem. However, it can also be observed that the proposed method significantly outperforms $\beta$-VAE on complex datasets (CelebA, Food101, CIFAR-10).
> > While many settings are the same, it is important to note that Table 3 and Table 5 have different objectives and premises. Therefore, a direct comparison of values might not be fair. On the other hand, in Table 5, a performance comparison with VAE-based methods is conducted. Since $\sigma$-VAE, has demonstrated its usefulness for $\beta$-VAE, it is not adopted as a comparison method in this experiment. However, considering the insights related to the results in Table 3, it could be interesting for readers to see a comparison between the best $\beta$-VAE and the proposed method in Table 5. When submitting the camera-ready version, we will consider adding a comparison experiment with the best $\beta$-VAE in Table 5. Thank you once again for your valuable feedback on the manuscript.
> >
> > **How are the hyper-parameters ... I was reviewing the paper.**
> > It might have been difficult to find the section on hyperparameters in the manuscript. As stated in the Settings, for all models, the number of latent units L is set to 16 for MNIST and to 64 for more complex datasets (CelebA, SVHN, Food101, CIFAR-10), following the hyperparameter settings in the original authors' papers. In the revised manuscript, we have made this clearer in the Appendix, so please check there for more details.
> > You might also be concerned about the impact of hyperparameters in the experiments shown in Table 5. In the experiments with the benchmark method, the $\sigma$-VAE, the FID for SVHN is reported to be 22.25 in the literature, which ourperforms the accuracy of the $\beta$-VAE. In our replication experiments with the $\sigma$-VAE, we achieved an FID of 18.44 (Table 5), indicating that our results are comparable or superior to those in the original paper. This confirms that the hyperparameter tuning was appropriately conducted. Furthermore, it is confirmed that the accuracy of our proposed method outperforms this level.
> >
> > **As you have incorporated ... additional time for the readers.**
> > Thank you for your advice on improving the manuscript. As you mentioned, it would be useful to include the details of the selected dissimilarity function in Table 5 to improve readability. We have corrected the problem by adding it to the caption of Table 5.
> >
> > I have revised the manuscript in accordance with your careful advice and would appreciate it if you could check it.

---

> > > ### Comment · Reviewer_MY4X · 2023-11-22
> > >
> > > Thanks for your clarification.
> > >
> > > I will verify these details. At a quick glance, some of my concerns are resolved.
> > >
> > > In comparison, I think the article is borderline. But I will keep the current score before discussing it with other reviewers. Further feedback will be given after the discussion and careful checking.

---

### Official Review · Reviewer_NNHM · 2023-10-30

**Soundness:** 2 fair
**Presentation:** 3 good
**Contribution:** 2 fair
**Rating:** 5
**Confidence:** 2

**Summary:**

This paper focused on improving variational autoencoder (VAE). Instead of adopting Gaussian settings, which is shown to violate domain-specific properties, this paper introduced a novel distribution family, i.e., the exponential dissimilarity-dispersion family (EDDF).  Correspondingly, an approximated algorithm using a log-expected dissimilarity loss is introduced to optimize VAE with the proposed EDDF decoders. Empirical validation is provided on a toy 1-D example. Effectiveness of the proposed model is evaluated on vision datasets.

**Strengths:**

**Originality:** this paper introduced a new distribution family for VAE. It also shows that some well-known distribution families can be interpreted as a subset of the new distribution family (EDDF). Correspondingly, an approximate algorithm is proposed for the training of VAE using the new EDDF decoder.

**Presentation:** this paper overall is well presented. The flow of the paper is smooth and easy to follow.

**Weaknesses:**

**Technical novelty:** the section 4 on VAE optimization is showing the derivations for the approximate optimization of VAE. However, its technical novelty is not clear to me. How does the proposed algorithm relate to or different from existing methods (e.g., Rybkin et al., 2021; Lin et al., 2019)?

Is there any theoretical guarantee or analysis to quantify the approximation? Simply using a 1-D toy model is not sufficient to valid the claim.

**Significance:** the significance of the paper is unclear to me.  1) Compared to other variants of VAE, the proposed approach is not achieving the best performance on all the datasets (Table 3). 2) how does the proposed model compare to other generative models (e.g., GAN)? Though the goal of the paper is to improve VAE, it is important to compare to other generative models to understand its significance.

**Presentation:** The term ‘domain knowledge’ used in the paper is not fitting the presentation well in my eyes.  How does the domain knowledge relate to the proposed EDDF decoder is not well defined at all.

**Questions:**

Please refer to the weakness section for my major concerns. In addition:

1.	what’s the computational cost of the proposed method?
2.	Can you give some insights or explanations on why the proposed model performs better on some of the datasets but worse on others in Table 3?
3.	Can you give any justifications on how the proposed distribution family encodes domain knowledge?

---

> ### Author Response · Authors · 2023-11-16
> **Response to Reviewer NNHM (1)**
>
> Thank you for reviewing the manuscript. We will answer your questions below. Please note that the 1-D validation is preliminary, and the main verifications in this paper are conducted using 2-D data.
>
> **Technical novelty: the section 4 on VAE optimization ... from existing methods (e.g., Rybkin et al., 2021; Lin et al., 2019)?**
> The novelty of our proposed method lies in extending the applicability of VAEs, widely used across various fields, by proposing an approach that maintains the validity of the probabilistic model. Specifically, as mentioned in Viewpoint (i) of the introduction, the introduction of a family of distributions is necessary to utilize prior information within VAEs that are already theoretically substantiated. Previous research has attempted to introduce additional mechanisms to balance reconstruction and regularization errors (e.g., Rybkin et al., 2021; Lin et al., 2019). However, these approaches did not consider the probabilistic framework inherent to the original VAE, making probabilistic interpretation challenging even when accuracy was improved. Therefore, this paper aims to resolve these traditional challenges by attempting to learn using domain knowledge in the form of dissimilarity functions while maintaining the probabilistic framework of VAEs. We propose a unified approach to balance reconstruction and regularization through variational inference with a probabilistic foundation provided by distribution families. Our experiments have verified that this approach performs as well or better than conventional ones. By establishing and demonstrating a theory that incorporates domain knowledge into VAEs through a probabilistically consistent approach, we have shown the potential to integrate insights from various fields into VAEs as inductive biases in a unified way. Given that VAEs are already used in various fields and practical challenges, we believe the application range of our proposed theory will broaden, and this paper constructs and validates the foundational theory for it. At this stage, the contribution of our theory remains within the domain of VAEs, but we believe it provides valuable information for the community in the field of generative modeling.
>
> **Is there any theoretical guarantee or analysis ... to valid the claim.**
> The main manuscript carries out validations on the quantification of approximations in a simple 1-D setting. Additionally, validations regarding the validity of γ<<1 are performed in appendix G of the original manuscript.
>
> **Significance: the significance of the paper is unclear to me. ... understand its significance.**
> Indeed, the experimental results in Table 3 do not always show our proposed method as superior. However, the technical contribution of this paper is not to achieve the highest precision in a specific task but to derive a theoretical framework for the introduction of any dissimilarity into VAEs. The focus of the experiments in Table 3 is to verify the validity of the newly proposed EDDF family of distributions and their estimation algorithm within the unified framework across various datasets. The fact that our proposed method, which features automatic adjustment of the scale parameter, exhibits performance comparable to the existing literature confirms the effectiveness of our theory. Meanwhile, as pointed out, discussions on the relationship and applicability to other generative models are necessary. Thus, the revised manuscript includes additions to this aspect.
>
> **Presentation: The term ‘domain knowledge’ used ... well defined at all.**
> In this paper, data characteristics are interpreted and discussed as distribution. An example of the relationship between distribution and EDDF decoder is shown in Table 1. The novelty of this paper is that it is possible to incorporate various data distribution assumptions in a unified framework based on probability theory. As you mentioned, the term "domain knowledge" is abstract and lacks direct explanation of the correspondence with the data distribution, so we have revised the manuscript to clarify.

---

> ### Author Response · Authors · 2023-11-16
> **Response to Reviewer NNHM (2)**
>
> **what’s the computational cost of the proposed method?**
> As originally described in the architecture details in the appendix, the computation cost is about 1h using a single GPU (RTX 2080Ti). Other parameters are also originally described in the appendix.
>
> **Can you give some insights or explanations on why the proposed model performs better on some of the datasets but worse on others in Table 3?**
> Using the visual domain as an example, this experiment comprehensively examines the dissimilarity functions proposed in the past, not limited to Neural Network-friendly similarity measures. Since the proposed method includes the aspect of gradient descent applicability as an implicit requirement, the applicability of the dissimilarity function may be quantified in terms of differentiability in the domain domain. Therefore, one drawback of the proposed method is that its performance is highly dependent on the Neural Network-Friendly dissimilarity function. The indicators of unstable performance are mainly those proposed before the rise of Deep Learning, and are considered to include operations that are not stochastic gradient descent-friendly. In such cases, it is considered necessary to introduce a more stable method of gradient calculation, as it may affect the stable learning of the proposed method. However, please note that neural network-based LPIPS and other methods are differentiable and thus provide stable learning. The above suggests that a differentiable data structure is required for the data domain, which is a limitation in the application of the proposed method.
>
> **Can you give any justifications on how the proposed distribution family encodes domain knowledge?**
> In this paper, we have demonstrated the effectiveness of the theory primarily through quantitative considerations. We believe that qualitative evaluation is a better way to understand the effectiveness of encoding domain knowledge. Figures 3 and 4 in the Appendix show the qualitative evaluation results corresponding to Tables 3 and 5. It can be seen that the proposed method is able to reflect the style and details of the input images better than other methods on relatively complex datasets such as Food-101 and CelebA. This indicates that the proposed method encodes knowledge important to the domain.
>
> If the explanations provided have resolved your concerns, we would appreciate it if you could reconsider the score given.

---

> ### Author Response · Authors · 2023-11-21
> **2 Days Left Reminder: Seeking Reviewer Feedback and Resolving Concerns for Reviewer NNHM**
>
> We appreciate your inquiry and thank you for your advice. We would like to make sure our response meets your needs. If you have any more questions or if there's anything in our response that doesn't fully address your concerns, please let us know. Your feedback is important to us and we'd be happy to discuss further. With just two days left in the discussion period, we're wondering if you've had a chance to review our response and if there's anything else we can help with.

---

> > ### Comment · Reviewer_NNHM · 2023-11-21
> > **Responses to the authors**
> >
> > Thank you for the reply and the reminder. My concerns remain. I am not convinced by the authors’ answers.
> >
> > Technical novelty: the authors emphasize the main novelty distinguishing the proposed work from existing works (e.g., Rybkin et al., 2021; Lin et al., 2019) lies in maintaining a probabilistic framework and using domain knowledge. The probabilistic framework is unclear to me.
> >
> > Theoretical justifications: The authors are only providing empirical justifications.
> >
> > Domain Knowledge:  For the domain knowledge, it should be something prior known, instead of being something implicit and only validated through empirical performance improvement.
> >
> > Based on the above remaining concerns, I will keep my original rating.

---

> > > ### Author Response · Authors · 2023-11-22
> > > **Response to Reviewer NNHM**
> > >
> > > Thank you for your constructive comments. We wholeheartedly agree with your remarks about the importance of theoretical guarantees and principles. Regarding novelty, as we have already explained, we will respond to your question about the validity of our approach. In this study, which focuses on natural observational data, empirical validation plays a crucial role. There are many instances where it is challenging to explicitly provide domain knowledge when dealing with natural observational data. However, we can assume distributions that are not solely Gaussian to a certain extent, based on empirical evidence. In such cases, we believe it is vital to construct a unified framework that allows for the introduction of various domain knowledge while maintaining the theories of previous research.
> > >
> > > Our study is based on this hypothesis and constructs a methodology accordingly. We believe that the validity of this approach needs to be supported by sufficient and rich empirical evidence. Thus, in our research, starting with the hypothesis of introducing domain knowledge into natural observational data, it is reasonable to assert validity through experimental observation. We hope this explanation helps you understand the process of our hypothesis testing.

---

> > > > ### Comment · Reviewer_NNHM · 2023-11-22
> > > > **Responses to the Author**
> > > >
> > > > Thank you for the explanation.  It clarifies the hypothesis and the authors’ motivations for using the term ‘domain knowledge’.  Considering a better wording or including additional justifications into the paper can help avoid confusion.
> > > >
> > > > I will wait for further reviewer discussion before modifying my score.

---

### Official Review · Reviewer_NsAA · 2023-10-30

**Soundness:** 2 fair
**Presentation:** 3 good
**Contribution:** 2 fair
**Rating:** 5
**Confidence:** 3

**Summary:**

In the paper, the authors formulate a method to incorporate domain knowledge into VAEs by specifying a dissimilarity function. This function is incorporated into the decoder's output distribution with proper normalization. Further, they reformulate the ELBO of the VAE based on the specified disimilarity function and provides analytical values for the normalizing constant in the limit $\gamma\rightarrow 0$.

**Strengths:**

A compact and straightforward loss function for VAEs with dissimilarity functions is presented. The methodology is easy to follow and the results are conclusive.

**Weaknesses:**

The paper's contribution is limited, as dissimilarity functions in the context of generative models have been presented in the past. The ELBO in (25) is probably the leading paper contribution, but beyond this point, I do not see any other contribution that justifies a whole paper around this idea.

**Questions:**

How do you extend this idea to other generative models such as GANs or latent diffusion models?

---

> ### Author Response · Authors · 2023-11-15
> **Response to Reviewer NsAA**
>
> Thank you for reviewing the manuscript. We will answer your questions below.
>
> **Weaknesses: The paper's contribution is limited, ... around this idea.**
> The novelty of our proposed method lies not just in suggesting a new dissimilarity function but in providing a unified framework for employing such functions within VAEs across various fields, along with a methodology for the automatic estimation of scale parameters. We have established a new theory that incorporates domain knowledge into VAEs in a probabilistically consistent manner, demonstrating the potential to integrate insights from various domains into VAEs as functional biases in a unified way.
> As VAEs are one of the representative generative models alongside GANs and diffusion models and are used in various fields and practical challenges, the application range of our proposed theory is expected to expand broadly. This paper constructs and validates the foundation of this new theory. Although the current contribution of this paper has not yet extended to direct applications in GANs or diffusion models, we believe it provides valuable insights for the deep learning and generative modeling communities.
>
> **Questions: How do you extend this idea to other generative models such as GANs or latent diffusion models?**
> VAEs, GANs, and diffusion models each offer distinct advantages as generative models. In this paper, we propose enhancements to the versatility of VAEs, which have a probabilistic foundation. Meanwhile, GANs and diffusion models are garnering attention for high-quality image generation based on large datasets and substantial model sizes. It is conceivable that considering data characteristics and domain knowledge will become important in these models as well, where this paper can make a contribution.
> Specifically, the stochastic processes inherent in diffusion models may offer a higher potential for applying the concepts of this theory. For instance, incorporating the concepts of this paper into the denoising process could be envisaged. We have added 'potential' as future work in the manuscript.
>
> If the explanations provided have resolved your concerns, we would appreciate it if you could reconsider the score given.

---

> ### Author Response · Authors · 2023-11-21
> **2 Days Left Reminder: Seeking Reviewer Feedback and Resolving Concerns for Reviewer NsAA**
>
> We appreciate your inquiry and thank you for your advice. We would like to make sure our response meets your needs. If you have any more questions or if there's anything in our response that doesn't fully address your concerns, please let us know. Your feedback is important to us and we'd be happy to discuss further. With just two days left in the discussion period, we're wondering if you've had a chance to review our response and if there's anything else we can help with.

---

> > ### Author Response · Authors · 2023-11-22
> > **Final Day Reminder: Discussion Period Closing Soon for Reviewer NsAA**
> >
> > We would like to remind you that, like the other reviewers, we are awaiting your valuable feedback. We understand that you are very busy, but we would appreciate it if you could respond before the deadline.

---

### Official Review · Reviewer_GpDS · 2023-10-31

**Soundness:** 3 good
**Presentation:** 3 good
**Contribution:** 2 fair
**Rating:** 6
**Confidence:** 4

**Summary:**

The paper considers the problem of training a VAE with "arbitrary" reconstruction loss, dubbed as a dissimilarity function. To do so, a Gibbs style distribution is introduced --- the Exponential Dissimilarity-Dispersion Family (EDDF) --- which is used to define the probabilistic reconstruction of a latent variable in the VAE. To make this tractable (in particular the corresponding normalization of the EDDF) several approximation are made yielding a simple log-expected reconstruction loss.

**Strengths:**

- The generalization of the standard VAE's reconstruction loss to that of a general dissimilarity function intuitively makes sense.
- The corresponding EDDF defined also follows celebrated patterns of exponential families and Gibbs-style distributions.
- The corresponding results are promising.

**Weaknesses:**

- I am a bit unconvinced why defining the EDDF is needed in the loss function of the VAE. (See questions below)
- Parts of the reasoning in Section 4 seems unclear. (See questions below)

**Questions:**

Why is EDDFs and defined explicitly?

- One aspect I am unclear about is why the reconstruction loss is not just defined directly as $\mathbb{E}[d(\textbf{x}, \hat{\textbf{x}}(\textbf{z}))]$ (expectation wrt ${q_{\phi}(\textbf{x}, \textbf{z})}$)? Even the log-expected reconstruction loss in Eq. (24) can be considered as an upper bound of taking log of the dissimilarity measure $d$, ie, $\log d(\textbf{x}, \hat{\textbf{x}}(\textbf{z}))$ (using Jensen's).
- The above perspective also skips the need for approximating the normalizer of the EDDF. I feel like this perspective can be made especially since the decoding is done deterministically in practice and thus an explicit density $p_{\mathbf{\theta}}(\textbf{x} \mid \textbf{z})$ does not seem to be needed. As such, additional input for why $p_{\mathbf{\theta}}(\textbf{x} \mid \textbf{z})$ should be defined explicitly would be great (and if there is prior work which took such a simplistic approach).
- One further note, the above perspective seems to fit in the narrative of "generalized variational inference" (See, eg, "An optimization-centric view on Bayes' rule: reviewing and generalizing variational inference" Section 4.4.4, switching log-likelihood for a general dissimilarity function).

Clarity in Section 4

- Eq. (20) is misleading as from Eq. (29) the equality should instead be an upper bound due to Jensen's.
- Why is the M / 2 factor not utilized in Eq. (24)?
- Above Eq. (25) there is a mention of "well-trained autoencoder" with a condition on $D$. I believe this should be $\gamma$ instead?

Other:

- It is worth clarifying what the regularization loss is in Eq. (25). I believe it is Eq. (7) with isotropic Gaussian (after digging through part of the code), but I am unsure given the current text.
- Something to add to Section 3.1: It might be worth mentioning that exponential family distributions can be explicitly be characterized as a density $ \propto \exp(- B_f(\cdot, \textbf{m}))$ where $ B_f$ is a Bregman divergence, see, eg, "Information Geometry and Its Applications" Section 2.7.

---

> ### Author Response · Authors · 2023-11-16
> **Response to Reviewer GpDS**
>
> Thank you very much for your careful review of the manuscript, **including checking the source code**. We also thank you for the many suggestions for improvement of the manuscript. The following are the answers to the questions we received.
>
> **One aspect I am unclear about is why the reconstruction loss is not just defined directly .. (using Jensen's).**
> In this paper, reconstruction loss was defined using a specific discrepancy measure d in order to capture the rich characteristics of the data in more detail. This measure captures the complexity of the data and allows for flexibility in the model. Using Jensen's inequality allows for more robust constraints to be applied to the loss function during the optimization process, resulting in more precise management of information transfer from latent variables to data variables.
>
> **The above perspective also skips the need for approximating ...  took such a simplistic approach).**
> Your point about not needing to approximate the normalization term in EDDF is correct. This approach was chosen to reduce computational costs and allow for practical applications. Indeed, for deterministic decoding, explicit densities are not required, but this may limit the versatility of the model. Therefore, we defined explicit densities to provide a more general framework applicable to different types of data.
>
> **One further note, the above perspective seems to ... general dissimilarity function).**
> The "generalized variational inference" perspective you have introduced is perfectly in line with our research motivation. From this perspective, our approach extends the framework of probabilistic variational inference to handle a wider range of discrepancy functions. This allows for better adaptability to specific data sets and tasks. Thank you for your deep insight.
>
> **Eq. (20) is misleading as from Eq. (29) the equality should instead be an upper bound due to Jensen's.**
> Thank you for pointing this out. We have clarified the text to avoid misunderstandings.
>
> **Why is the M / 2 factor not utilized in Eq. (24)?**
> This was a typographical error and has been corrected. The implementation has been reconfirmed as correct.
>
> **Above Eq. (25) there is a mention of "well-trained autoencoder" ..  instead?**
> The wording has been revised and clarified.
>
> **It is worth clarifying what the regularization loss is in Eq. (25). ... the current text.**
> The wording has been revised and clarified.
>
> **Something to add to Section 3.1: ...  Section 2.7.**
> Thank you very much for your helpful comments. We have added them to the manuscript.
>
> Please note that some of the comments we have received are beyond our knowledge. We would appreciate your advice if the corrections are not appropriate. If the above has answered your questions, we would be happy to consider reviewing the score.

---

> > ### Comment · Reviewer_GpDS · 2023-11-19
> >
> > Thank you for the response and revised manuscript.
> > In terms of my comments, they have been appropriately answered.
> >
> > I'll be waiting for other reviewers to comment about the rebuttal and further reviewer discussion before modifying my score.

---

> > > ### Author Response · Authors · 2023-11-20
> > > **Thank You for Confirming the Revised Manuscript**
> > >
> > > We are deeply grateful for your time in reviewing our revised manuscript and providing insightful comments. We eagerly anticipate a fruitful and engaging discussion. Thank you very much for your esteemed cooperation.

---

### Official Review · Reviewer_xCRZ · 2023-11-08

**Soundness:** 2 fair
**Presentation:** 2 fair
**Contribution:** 2 fair
**Rating:** 5
**Confidence:** 3

**Summary:**

The paper is concerned with the possibility to replace the normal distribution in the VAE framework by a distribution from some family that includes a dissimilarity function and a dispersion parameter. The paper seems to propose minimizing the ELBO objective, contrary to the usual maximization, and claims that a decoder with such a distribution can induce arbitrary dissimilarity functions as the reconstruction loss. Some empirical checks of the method, on simple datasets in visual domain, are described.

**Strengths:**

Paper is devoted to an exploration of possibilities for modifications of widely used VAE method

**Weaknesses:**

1) The writing of the paper is somewhat obscure, especially in what concerns long sentences, cf below.

2) The rational to consider the minimization, instead of the standard maximization, of the ELBO objective is not clear.

3) The results seems to be too incremental, many previous works have proposed to replace the Gaussian distribution in VAE by some other distributions.

4) In  2 out of 5 cases in the  described empirical results, the baselines from previous papers performed better (Table 5)

5) The standard deviations in the experiments are not reported.

6) Only two out of many metrics evaluating VAE  quality are calculated.  In particular, the reconstruction errors are not reported in the comparison tables.

7) Rather simple datasets are considered only. Also only the visual domain is considered.

Some remarks :

page 1 : "VAEs with the commonly used settings practically suffer from performance
degradation, such as reconstruction fidelity and generation naturalness" -> perhaps ... insufficient reconstruction fidelity and... Since as it stands, both the "reconstruction fidelity" and "generation naturalness" appear as undesired/bad qualities.

pages 2-3: "...with negative ELBO objective... whose maximization ..." - usually the negative ELBO objective is minimized

page 4, last equation in (16) - it seems that the paper's method seeks to maximize the reconstruction loss, given by a positive distance cf eqs (9), (18).

Following improvements made during rebuttal I'm raising the score, however some issues require further work.

**Questions:**

Some suggestions: To demonstrate the effectiveness of the method, the  standard deviations should be measured in the experiments for various random seeds. Also, more than just two metrics should be measured. The comparison with baselines performance in other domains and on more complex real- world datasets is also necessary for evaluation of the method's practicality.

---

> ### Author Response · Authors · 2023-11-15
> **Response to Reviewer xCRZ (1)**
>
> We apologize for a word error in our manuscript that appears to have led to a misunderstanding. In the explanation of the general structure of VAEs, the term "maximization" under Equation (1) should have been "minimization". As you correctly pointed out, the negative ELBO is indeed the subject to be minimized, and our treatment of ELBO optimization in VAEs in this paper is consistent with standard practice. This section pertains to the general formalization of VAEs and is not directly related to our proposed method. Given that the presentation scores from the other four reviewers were all highest, we believe that this single word error has not significantly impacted the overall manuscript. Moreover, the novelty of our proposed method is described in Sec. 3.
>
> **As we mentioned in the introduction, our novelty lies not in the minimization of ELBO but in proposing a framework that maintains the probabilistic model structure while incorporating domain knowledge into the optimization of ELBO, and we kindly request a review focused on the novel aspects.** Thank you for pointing out this important issue. We are grateful for your meticulous and constructive review.

---

> ### Author Response · Authors · 2023-11-15
> **Response to Reviewer xCRZ (2)**
>
> **The writing of the paper is somewhat obscure, especially in what concerns long sentences, cf below.**
> Thank you for your feedback. We have revisited the entire manuscript and clarified the sentence accordingly.
>
> **The rational to consider the minimization, instead of the standard maximization, of the ELBO objective is not clear.**
> This was a word error in the explanation of the general introduction of VAEs. We apologize for any confusion caused.
>
> **The results seems to be too incremental, many previous works have proposed to replace the Gaussian distribution in VAE by some other distributions.**
> In this paper, we present not merely a substitution of simple distributions but a unified framework for dissimilarity functions in VAEs, along with methodologies to realize it while maintaining the validity of the probabilistic model. The differences between prior work and our proposed approach are primarily detailed in the related work section of the original manuscript; please refer to it. For instance, [Czolbe et al., "A Loss Function for Generative Neural Networks Based on Watson’s Perceptual Model," 2020](https://papers.nips.cc/paper/2020/hash/165a59f7cf3b5c4396ba65953d679f17-Abstract.html) and [Esser et al., "Taming Transformers for High-Resolution Image Synthesis," 2021](https://compvis.github.io/taming-transformers/) employ perceptual loss. However, these maintain a constant scale in the distribution, which fails to leverage the Bayesian properties of VAEs, leading to a balancing problem where the decoder cannot capture uncertainty. Moreover, these perceptual losses are introduced in an ad-hoc manner, making it challenging to tie them to the rich theoretical foundation that VAEs possess.
> Our proposed method suggests a versatile dissimilarity function that preserves the theoretical coherence inherent in VAEs, allowing for the introduction of any function deemed significant for the data domain under consideration, as discussed in viewpoint II. Also, while [Takida et al., "Preventing oversmoothing in VAE via generalized variance parameterization," 2022](https://www.sciencedirect.com/science/article/pii/S0925231222010591) is mentioned as prior work, their consideration is limited to Gaussian cases, making it difficult to incorporate data domain-specific dissimilarity functions.
>
> **In 2 out of 5 cases in the described empirical results, the baselines from previous papers performed better (Table 5)**
> Indeed, the results in Table 5 do not always favor our proposed method. However, the technical contribution of this paper is not about achieving the highest precision in image generation tasks, but about the theoretical derivation and validation of a unified framework for introducing arbitrary dissimilarity into VAEs. Therefore, the most notable aspect of our experiments is the verification of whether the EDDF family of distributions and its estimation algorithm operate validly. The fact that our proposed method performs comparably to existing literature validates the effectiveness of our theory.
>
> **The standard deviations in the experiments are not reported. Only two out of many metrics evaluating VAE quality are calculated. In particular, the reconstruction errors are not reported in the comparison tables.**
> The primary focus of our experiments is to demonstrate a theory that incorporates domain knowledge into the probabilistic model characteristics while preserving its properties. To validate this theory, it is necessary to comprehensively evaluate dissimilarity functions proposed in the past. Using these dissimilarity metrics directly as evaluation criteria would lead to an unfair assessment and could shift the focus away from the main objective of the verification, potentially causing confusion. Therefore, we uniformly employed the current metrics for this purpose. Consequently, we believe that FID and KID are suitable unified metrics for the evaluations in this verification and have adopted them. Furthermore, we have validated the effectiveness of our approach across a broader range of datasets compared to previous literature. We will address the standard deviation below.
>
> **Rather simple datasets are considered only. Also only the visual domain is considered.**
> The primary application area for VAEs is the image domain, which also readily lends itself to the expression of domain knowledge, hence we have conducted validations using major image datasets. From the perspective of incorporating domain knowledge, considering the characteristics of the data is crucial. We believe that evaluations with five types of datasets are extensive compared to related prior research.

---

> ### Author Response · Authors · 2023-11-15
> **Response to Reviewer xCRZ (3)**
>
> **page 1 : "VAEs with the commonly used settings ... appear as undesired/bad qualities.**
> Thank you for your comment. It has been corrected.
>
> **pages 2-3: "...with negative ELBO objective... objective is minimized. page 4, last equation ... eqs (9), (18).**
> The error regarding minimization has been rectified.
>
> **Some suggestions:**
> Thank you for your feedback. Indeed, we agree that conducting experiments with different random seeds is important for enhancing the reliability of the paper. As for the validity of the evaluation metrics, as stated above, we have utilized metrics suited for verifying our theory and selected for evaluating various dissimilarity functions, which we believe are appropriate for testing the effectiveness of the theory. Regarding the datasets, the selection is crucial for demonstrating our theory; thus, we have already performed validations with five different datasets, which is more extensive than in previous literature. Consequently, while we concur with the importance of experiments to assess the impact of random seeds in the camera-ready version, we cannot agree that it directly leads to a lower evaluation without considering other validations.
>
>
> If the explanations provided have resolved your concerns, we would appreciate it if you could reconsider the score given.

---

> > ### Comment · Reviewer_xCRZ · 2023-11-23
> >
> > Authors has rectified the errors in the descriptions of the objectives. The efforts towards the improvement of the text are appreciated, however I believe some issues including issues raised by other reviewers remain and the paper still needs another round of reviews.

---

> > > ### Author Response · Authors · 2023-11-23
> > > **Thank you for your feedback**
> > >
> > > Thank you very much for reviewing the revised manuscript. We completely agree with your opinion. Thank you for your valuable advice. We would be grateful if you could evaluate the contribution of this paper at the next review step.

---

> ### Author Response · Authors · 2023-11-21
> **2 Days Left Reminder: Seeking Reviewer Feedback and Resolving Concerns for Reviewer xCRZ**
>
> We appreciate your inquiry and thank you for your advice. We would like to make sure our response meets your needs. If you have any more questions or if there's anything in our response that doesn't fully address your concerns, please let us know. Your feedback is important to us and we'd be happy to discuss further. With just two days left in the discussion period, we're wondering if you've had a chance to review our response and if there's anything else we can help with.

---

> > ### Author Response · Authors · 2023-11-22
> > **Final Day Reminder: Discussion Period Closing Soon for Reviewer xCRZ**
> >
> > We would like to remind you that, like the other reviewers, we are awaiting your valuable feedback. We understand that you are very busy, but we would appreciate it if you could respond before the deadline.

---

### Author Response · Authors · 2023-11-16
**For all Reviewers**

Thank you very much for taking the time to review our manuscript amidst your busy schedule. In this paper, we propose the construction and formalization of a new theory based on probability theory aimed at improving the versatility of VAEs.
The main revisions to the manuscript are as follows:
- Corrected a descriptive error regarding the general explanation of VAEs.
- Corrected a notation error in Equation (24) (the code has been verified to be correct).
- Added details on the contributions of this paper and its applicability, including other generative models.

The revisions in the manuscript are indicated in blue. Lastly, we would like to express our gratitude once again for your careful and constructive reviews.

---

### Meta-Review · Area_Chair_bzPc · 2023-12-04

**Metareview:**

This paper proposes to change the VAE decoder to an exponential dissimilarity-dispersion family (EDDF), as described by eq.(16). The author gave an approximation of the resulting ELBO, which includes a dispersion parameter that can be automatically learned. The proposed method is tested on typical image datasets and the performance is on par with SOTA alternatives.

All reviewers have participated in the final discussion with necessary meeting arranged.

*Strength*:

The paper is well integrated with both theoretical parts and experiments, and the writing is easy to follow.

*Weakness*:

- The novelty is limited. VAE on exponential families and location-scale families are studied in the literature. The authors should make more efforts and be more modest to establish their standpoint from the literature. Theoretically, the author made an approximation of ELBO. This could be supported by further analysis of the quality of the approximation. There is no analytical result to support the authors' claim on a theoretical framework. Practically, the experimental results do not demonstrate the superiority of the proposed approach.

Below are further comments, only for the authors' future revision:

sec 3. Mention location-scale family. What are the requirements of $d$ to make $f_d$ a proper density?

sec 4. eq.(24) is introduced abruptly. What is the relationship with eq.(18)?

sec 4. Here, the approximation of ELBO should be more carefully analyzed, ideally with some results on the approximation error.

**Justification For Why Not Higher Score:**

As mentioned in the metareview, this work needs further development, either on the theoretical side or the practical side, to have enough significance. This work needs to be carefully compared with peers to establish a novel standpoint.

**Justification For Why Not Lower Score:**

N/A

---

### Decision · Program_Chairs · 2024-01-16

Reject